# Todyformer: Towards Holistic Dynamic Graph Transformers with Structure-Aware Tokenization

**Mahdi Biparva**[*]                           *mahdi.biparva@huawei.com*
*Huawei Noah's Ark Lab*

**Raika Karimi**[*]                             *raika.karimi@huawei.com*
*Huawei Noah's Ark Lab*

**Faezeh Faez**                                *faezeh.faez@huawei.com*
*Huawei Noah's Ark Lab*

**Yingxue Zhang**                             *yingxue.zhang@huawei.com*
*Huawei Noah's Ark Lab*

**Reviewed on OpenReview:** *https://openreview.net/forum?id=nAQSUqEspb*

## Abstract

Temporal Graph Neural Networks have garnered substantial attention for their capacity to model evolving structural and temporal patterns while exhibiting impressive performance. However, it is known that these architectures are encumbered by issues that constrain their performance, such as over-squashing and over-smoothing. Meanwhile, Transformers have demonstrated exceptional computational capacity to effectively address challenges related to long-range dependencies. Consequently, we introduce Todyformer—a novel Transformer-based neural network tailored for dynamic graphs. It unifies the local encoding capacity of Message-Passing Neural Networks (MPNNs) with the global encoding of Transformers through i) a novel patchifying paradigm for dynamic graphs to improve over-squashing, ii) a structure-aware parametric tokenization strategy leveraging MPNNs, iii) a Transformer with temporal positional-encoding to capture long-range dependencies, and iv) an encoding architecture that alternates between local and global contextualization, mitigating over-smoothing in MPNNs. Experimental evaluations on public benchmark datasets demonstrate that Todyformer consistently outperforms the state-of-the-art methods for downstream tasks. Furthermore, we illustrate the underlying aspects of the proposed model in effectively capturing extensive temporal dependencies in dynamic graphs. The code is publicly available at https://github.com/huawei-noah/noah-research/tree/master/graph_atlas

## 1 Introduction

Dynamic graphs, driven by the surge of large-scale structured data on the internet, have become pivotal in graph representation learning. Dynamic graphs are simply static graphs where edges have time attributes (Kazemi et al., 2020). Representation learning approaches for dynamic graphs aim to learn how to effectively encode recurring structural and temporal patterns for node-level downstream tasks. For instance, Future Link Prediction (FLP) uses past interactions to predict future links, while Dynamic Node Classification (DNC) focuses on predicting labels of upcoming nodes based on impending interactions (Xu et al., 2019a). While models based on Message-Passing Neural Networks (MPNN) (Gilmer et al., 2017; Luo et al., 2021) have demonstrated impressive performance on encoding dynamic graphs (Rossi et al., 2020; Wang et al., 2021; Jin et al., 2022; Luo & Li, 2022), many approaches have notable limitations. Primarily, these methods often rely

---

[*]equal contribution.

heavily on chronological training or use complex memory modules for predictions (Kumar et al., 2019; Xu et al., 2020; Rossi et al., 2020; Wang et al., 2021), leading to significant computational overhead, especially for dynamic graphs with many edges. Additionally, the use of inefficient message-passing procedures can be problematic, and some methods depend on computationally expensive random-walk-based algorithms (Wang et al., 2021; Jin et al., 2022). These methods often require heuristic feature engineering, which is specifically tailored for edge-level tasks.

Moreover, there is a growing consensus within the community that the message-passing paradigm is inherently constrained by the hard inductive biases imposed by the graph structure (Kreuzer et al., 2021). A central concern with conventional MPNNs revolves around the over-smoothing problem stemmed from the exponential growth of the model's computation graph (Dwivedi & Bresson, 2020). This issue becomes pronounced when the model attempts to capture the higher-order long-range aspects of the graph structure. Over-smoothing hurts model expressiveness in MPNNs where the network depth grows in an attempt to increase expressiveness. However, the node embeddings tend to converge towards a constant uninformative representation. This serves as a reminder of the lack of flexibility observed in early recurrent neural networks used in Natural Language Processing (NLP), especially when encoding lengthy sentences or attempting to capture long-range dependencies within sequences (Hochreiter & Schmidhuber, 1997). However, Transformers have mitigated these limitations in various data modalities (Vaswani et al., 2017; Devlin et al., 2018; Liu et al., 2021; Dosovitskiy et al., 2020; Dwivedi & Bresson, 2020). Over-squashing is another problem that message-passing networks suffer from since the amount of local information aggregated repeatedly increases proportionally with the number of edges and nodes (Hamilton, 2020; Topping et al., 2021).

MPNN-based models for learning dynamic graphs do not deviate from aforementioned drawbacks. To address these challenges, we propose Todyformer[1]—a novel Graph Transformer model on dynamic graphs that unifies the local and global message-passing paradigms by introducing patchifying, tokenization, and encoding modules that collectively aim to improve model expressiveness through alleviating over-squashing and over-smoothing in a systematic manner. To mitigate the neighborhood explosion (i.e, over-squashing), we employ temporal-order-preserving patch generation, a mechanism that divides large dynamic graphs into smaller dynamic subgraphs. This approach breaks the larger context into smaller subgraphs suitable for local message-passing, instead of relying on the model to directly analyze the granular and abundant features of large dynamic graphs.

Moreover, we adopt a hybrid approach to successfully encode the long-term contextual information, where we use MPNNs for tasks they excel in, encoding local information, while transformers handle distant contextual dependencies. In other words, our proposed architecture adopts the concept of learnable structure-aware tokenization, reminiscent of the Vision Transformer (ViT) paradigm (Dosovitskiy et al., 2020), to mitigate computational overhead. Considering the various contributions of this architecture, Todyformer dynamically alternates between encoding local and global contexts, particularly when capturing information for anchor nodes. This balances between the local and global computational workload and augments the model expressiveness through the successive stacking of the MPNN and Transformer modules.

## 2  Related Work

**Representation learning for dynamic graphs:** Recently, the application of machine learning to Continuous-Time Dynamic Graphs (CTDG) has drawn the attention of the graph community (Kazemi et al., 2020). RNN-based methods such as JODIE (Divakaran & Mohan, 2020), Know-E (Trivedi et al., 2017), and DyRep (Trivedi et al., 2019) typically update node embeddings sequentially as new edges arrive. STAR (Xu et al., 2019b) is one of the early works that uses attentive approach for temporal graphs. TGAT (Xu et al., 2020), akin to GraphSAGE (Hamilton et al., 2017) and GAT (Veličković et al., 2018), uses attention-based message-passing to aggregate messages from historical neighbors of an anchor node. TGN (Rossi et al., 2020) augments the message-passing with an RNN-based memory module that stores the history of all nodes with a memory overhead. CAW (Wang et al., 2021) and NeurTWs (Jin et al., 2022) abandon the common message-passing paradigm by extracting temporal features from temporally-sampled causal walks. TRRN (Xu et al., 2021) is another memory-based model that benefit from self-attention to reason over set of

---

[1]We are going to open-source the code upon acceptance.

memories. CAW operates directly within link streams and mandates the retention of the most recent links, eliminating the need for extensive memory storage. Moreover, Souza et al. (2022) investigates the theoretical underpinnings regarding the representational power of dynamic encoders based on message-passing and temporal random walks. DyG2Vec (Alomrani et al., 2022) proposes an efficient attention-based encoder-decoder MPNN that leverages temporal edge encoding and window-based subgraph sampling to regularize the representation learning for task-agnostic node embeddings. GraphMixer (Cong et al., 2023) simplifies the design of dynamic GNNs by employing fixed-time encoding functions and leveraging the MLP-Mixer architecture (Tolstikhin et al., 2021).

**Graph Transformers:** Transformers have been demonstrating remarkable efficacy across diverse data modalities (Vaswani et al., 2017; Dosovitskiy et al., 2020). The graph community has recently started to embrace them in various ways (Dwivedi & Bresson, 2020). Graph-BERT (Zhang et al., 2020) avoids message-passing by mixing up global and relative scales of positional encoding. Kreuzer et al. (2021) proposes a refined inductive bias for Graph Transformers by introducing a soft and learnable positional encoding (PE) rooted in the graph Laplacian domain, signifying a substantive stride in encoding low-level graph structural intricacies. Ying et al. (2021) is provably more powerful than 1-WL; it abandons Laplacian PE in favor of spatial and node centrality PEs. Subsequently, SAT (Chen et al., 2022) argues that Transformers with PE do not necessarily capture structural properties. Therefore, the paper proposes applying GNNs to obtain initial node representations. Graph GPS (Rampášek et al., 2022) provides a recipe to build scalable Graph Transformers, leveraging structural and positional encoding where MPNNs and Transformers are jointly utilized to address over-smoothing, similar to SAT. TokenGT (Kim et al., 2022) demonstrates that standard Transformers, without graph-specific modifications, can yield promising results in graph learning. It treats nodes and edges as independent tokens and augments them with token-wise embeddings to capture the graph structure. He et al. (2023) adapts MLP-Mixer (Tolstikhin et al., 2021) architectures to graphs, partitioning the input graph into patches, applying GNNs to each patch, and fusing their information while considering both node and patch PEs.

While the literature adapts Transformers to static graphs, a lack of attention is eminent on dynamic graphs. In this work, we strive to shed light on such adaptation in a principled manner and reveal how dynamic graphs can naturally benefit from a unified local and global encoding paradigm.

## 3 Todyformer: Tokenized Dynamic Graph Transformer

We begin this section by presenting the problem formulation of this work. Next, we provide the methodological details of the Todyformer architecture along with its different modules.

**Problem Formulation:** A Continuous-Time Dynamic Graph (CTDG) $\mathcal{G} = (\mathcal{V}, \mathcal{E}, \mathcal{X}^E, \mathcal{X}^v)$ with $N = |\mathcal{V}|$ nodes and $E = |\mathcal{E}|$ edges can be represented as a sequence of interactions $\mathcal{E} = \{e_1, e_2, \ldots, e_E\}$, where $\mathcal{X}^v \in \mathbb{R}^{N \times D^V}$ and $\mathcal{X}^E \in \mathbb{R}^{E \times D^E}$ are the node and edge features, respectively. $D^V$ and $D^E$ are the dimensions of the node and edge feature space, respectively. An edge $e_i = (u_i, v_i, t_i, m_i)$ links two nodes $u_i, v_i \in \mathcal{V}$ at a continuous timestamp $t_i \in \mathbb{R}$, where $m_i \in \mathcal{X}^E$ is an edge feature vector. Without loss of generality, we assume that the edges are undirected and ordered by time (i.e., $t_i \leq t_{i+1}$). A temporal sub-graph $\mathcal{G}_{ij}$ is defined as a set consisting of all the edges in the interval $[t_i, t_j]$, such that $\mathcal{E}_{ij} = \{e_k \mid t_i \leq t_k < t_j\}$. Any two nodes can interact multiple times throughout the time horizon; therefore, $\mathcal{G}$ is a multi-graph. Following DyG2Vec (Alomrani et al., 2022), we adopt a window-based encoding paradigm for dynamic graphs to control representation learning and balance the trade-off between efficiency and accuracy according to the characteristics of the input data domain. The parameter $W$ controls the size of the window for the input graph $\mathcal{G}_{ij}$, where $j = i + W$. For notation brevity, we assume the window mechanism is implicit from the context. Hence, we use $\mathcal{G}$ as the input graph unless explicit clarification is needed.

Based on the downstream task, the objective is to learn the weight parameters $\theta$ and $\gamma$ of a dynamic graph encoder $\text{ENCODER}_\theta$ and decoder $\text{DECODER}_\gamma$ respectively. $\text{ENCODER}_\theta$ projects the input graph $\mathcal{G}$ $\text{DECODER}$ to the node embeddings $\boldsymbol{H} \in \mathbb{R}^{N \times D^H}$, capturing temporal and structural dynamics for the nodes. Meanwhile, a decoder $\text{DECODER}_\gamma$ outputs the predictions given the node embeddings for the downstream task, enabling accurate future predictions based on past interactions. More specifically:

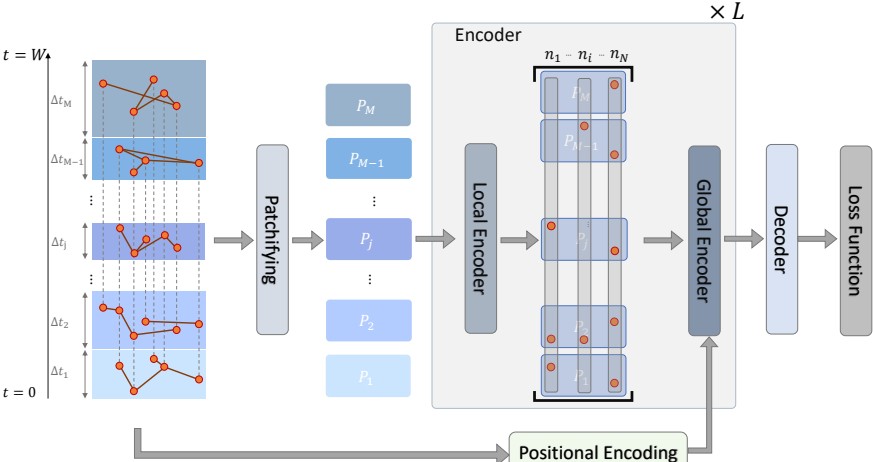

Figure 1: Illustration of Todyformer encoding-decoding architecture. Before being fed into the Encoder, the temporal graph is segmented into $M$ non-overlapping subgraphs (patches) based on the timestamps of the edges, with each subgraph containing an equal number of edges. This partitioning process divides the overall time interval into $M$ segments such that $W = \sum_{k=1}^{M} \Delta t_k$. Here, $W$ denotes the window size, $P_j$ represents the $j$-th patch, $n_i$ denotes the $i$-th node of the graph, and $L$ represents the number of encoder blocks.

$$\boldsymbol{H} = \text{ENCODER}_\theta(\mathcal{G}), \qquad \boldsymbol{Z} = \text{DECODER}_\gamma(\boldsymbol{H}), \tag{1}$$

Here, $\boldsymbol{Z}$ represents predictions for the ground-truth labels. In this work, we focus on common downstream tasks defined similarly to Alomrani et al. (2022) for training and evaluation: i) Future Link Prediction (FLP) and ii) Dynamic Node Classification (DNC).

### 3.1 Encoder Architecture

Todyformer consists of $L$ blocks of encoding $\text{ENCODER}_\theta = \{(\text{LOCALENCODER}^l, \text{GLOBALENCODER}^l)\}_{l=0}^L$ where $\text{LOCALENCODER} = \{\text{LOCALENCODER}^l\}_{l=0}^L$ and $\text{GLOBALENCODER} = \{\text{GLOBALENCODER}^l\}_{l=0}^L$ are the sets of local and global encoding modules, respectively. As illustrated in Figure 1, the encoding network of Todyformer benefits from an alternating architecture that alternates between local and global message-passing. The local encoding is structural and temporal, according to the learnable local encoder, and the global encoding in this work is defined to be temporal. We leave the structural and temporal global encoding for future work. In the following, we define each encoding module in more detail.

### 3.2 Patch Generation

Inspired by Dosovitskiy et al. (2020), Todyformer begins by partitioning a graph into $M$ subgraphs, each containing an equal number of edges. This partitioning is performed based on the timestamp associated with each edge. Specifically, the patchifier PATCHIFIER evenly segments the input graph $\mathcal{G}$ with $\mathcal{E} = \{e_1, e_2, \ldots, e_E\}$ edges into $M$ non-overlapping subgraphs of equal size, referred to as patches. More concretely:

$$\mathcal{P} = \text{PATCHIFIER}(\mathcal{G}; M) \tag{2}$$

where $\mathcal{P} = \{\mathcal{G}_m \,|\, m \in \{1, 2, ..., \frac{E}{M}\}\}$ and the $m$-th graph, denoted as $\mathcal{G}_m$, consists of edges with indices in the range $\{(m-1)\frac{E}{M} + 1, \cdots, m\frac{E}{M}\}$. Partitioning the input graph into $M$ disjoint subgraphs helps message-passing to be completely separated within each patch. Additionally, $M$ manages the trade-off between alleviating over-squashing and maintaining the local encoder's expressiveness. Through ablation studies, we empirically reveal how different datasets react to various $M$ values.

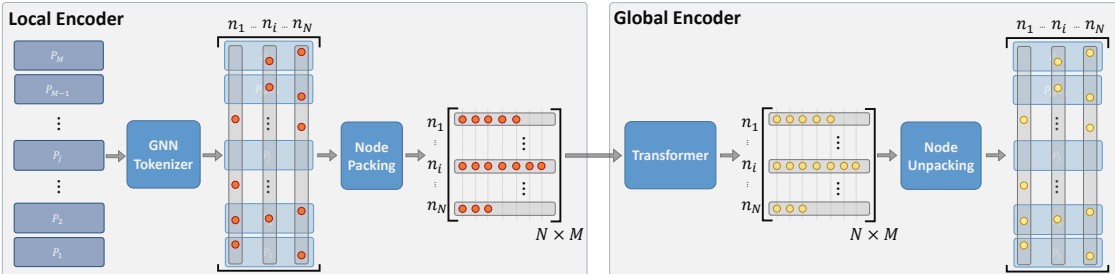

Figure 2: Schematic depiction of the computation flow in the local and global encoding modules, particularly highlighting node packing and unpacking modules in Todyformer.

### 3.3 Local Encoding: Structure-Aware Tokenization

Local encoding $\text{LOCALENCODER}^l = (\text{TOKENIZER}^l, \text{PACKER}^l)$ contains two modules: the tokenization $\text{TOKENIZER}^l$ and the packing $\text{PACKER}^l$ modules. The former handles local tokenization, and the latter packs tokens into a sequential data structure that will be consumed by the global encoder.

**Structure-Aware Tokenization:** Following the recent trend in Graph Transformers, where tokenization is structure-aware, local encoding in Todyformer utilizes a dynamic GNN to map the input node embeddings to the latent embeddings that a Transformer will process later on. It should be noted that the local encoder has learnable parameters to encode both structural and temporal patterns in the patches. Without loss of generality, we use DyG2Vec (Alomrani et al., 2022) as a powerful attentive message-passing model to locally encode input features into semantically meaningful node tokens.

$$\mathcal{H}^l = \text{TOKENIZER}^l(\bar{\mathcal{H}}^{l-1}) \tag{3}$$

where $\mathcal{H}^l = \{H_i^l\}_{i=0}^{M-1}$ is the set of node embeddings $H_i$ for $M$ different patches, and $\bar{\mathcal{H}}^{l-1}$ is the set of node embeddings computed by the previous block. As illustrated in Figure 1, the output of one block from the global encoder is transferred as the input into the local encoder of the subsequent block. It should be noted that $\bar{\mathcal{H}}^0 = \mathcal{X}$ for the first layer, where $\mathcal{X} = \{X_i\}_{i=0}^{M-1}$ is the set of node features for all patches.

**Packing:** Once the node features are locally encoded into node tokens, the next step is to pack the set of node embeddings $\mathcal{H}^l$ into the standard format required by Transformers. Since a node may appear in multiple patches, to collect all the node embeddings for a particular node across the patches, a node-packing module $\text{PACKER}^l$ is utilized. This module collects the embeddings of all nodes across the patches and arranges them in a sequential data format as follows:

$$H^l = \text{PACKER}^l(\mathcal{H}^l, \mathcal{P}) \tag{4}$$

where $H^l \in \mathbb{R}^{N \times M \times D^H}$ such that $N$ is the number of nodes in the input graph $\mathcal{G}$, $M$ is the total number of patches, and $D^H$ is the dimension of the embedding space. The module $\text{PACKER}^l$ uses $\mathcal{P}$ to figure out which patches a node belongs to. Consequently, the output of the local encoding module is structured in a tensor that can be easily consumed by a Transformer. The computation flow in the local encoder is shown in Figure 2. Since nodes may have interactions for a variable number of times in the input graph, it is necessary to pad the short sequences with the [MASK] tokens at the end. Then, the mini-batch of sequences can be easily packed into a dense tensor and fed as input to Transformers.

### 3.4 Global Encoding

The packed node tokens will be fed into the global encoding module to perform long-range message-passing beyond the local context of the input patches. Therefore, Todyformer not only maximizes

the parametric capacity of MPNNs to encode local context but also leverages the long-range capacities of Transformers to improve the model expressiveness. The global encoder $\text{GlobalEncoder}^l = (\text{PositionalEncoder}^l, \text{Trans}^l, \text{Unpacker}^l)$ consists of the positional encoder $\text{PositionalEncoder}^l$, Transformer $\text{Trans}^l$, and unpacking module $\text{Unpacker}^l$ according to the details provided in the following.

**Positional Encoding:** Transformers are aware of the ordering in the input sequences through positional encoding. Various systematic approaches have been investigated in the literature for the sake of improved expressiveness (Dwivedi & Bresson, 2020; Kreuzer et al., 2021). Once the structural and temporal features are locally mapped into node tokens, and the sequential input $H^l$ is packed at layer $l$, positional encoding is needed to inform the Transformer of the temporal ordering of the node tokens on a global scale. The positional encoding in Todyformer is defined based on the notion of the position and the encoding function. The position can be explicitly defined as the global edge index of a node upon an interaction at a timestamp or implicitly defined as the patch or occurrence indices. The encoding function can be a linear or sinusoidal mapping. The positional encoding P is fused into the packed node embeddings through the addition modulation, as follows:

$$H^l = H^l + P, \qquad P = \text{PositionalEncoder}^l(\mathcal{P}) \in \mathbb{R}^{N \times M \times D^H} \tag{5}$$

**Transformer:** The global encoding updates node embeddings through a dot-product Multi-head Self-Attention (MSA) Transformer architecture as follows: The global encoder computes a contextualized embedding for each node based on the representation learned for that node across different patches. This is accomplished by utilizing a Transformer architecture as follows:

$$\bar{H}^l = \text{Trans}^l(H^l), \qquad \text{Trans}^l = \texttt{Transformer}(Q, K, V) = \text{softmax}\Big(\frac{QK^T}{\sqrt{d_k}}\Big)V \tag{6}$$

where $Q = H^l W_q \in \mathbb{R}^{N \times M \times D^k}$, $K = H^l W_k \in \mathbb{R}^{N \times M \times D^k}$, and $V = H^l W_v \in \mathbb{R}^{N \times M \times D^v}$ are the query, key, and value, respectively, and $W_q$, $W_k \in \mathbb{R}^{D^H \times D^k}$ and $W_v \in \mathbb{R}^{D^H \times D^v}$ are learnable matrices, which are shared among all the graph nodes. We apply an attention mask to enforce directed connectivity between node tokens through time, where a node token from the past is connected to all others in the future. The Transformer module is expected to learn temporal inductive biases from the context on how to deploy attention on recent interactions versus early ones.

**Unpacking:** For intermediate blocks, the unpacking module $\text{Unpacker}^l$ is necessary to transform the packed, unstructured sequential node embeddings back into the structured counterparts that can be processed alternately by the local encoder of the next block. It is worth mentioning that the last block $L$ does not require an unpacking module. Instead, a readout function $\text{Readout}$ is defined to return the final node embeddings consumed by the task-specific decoding head:

$$\bar{\mathcal{H}}^l = \text{Unpacker}^l(\bar{H}^l), \qquad \bar{H}^L = \text{Readout}(\bar{H}^{L-1}) \in \mathbb{R}^{N \times D^H} \tag{7}$$

where $\bar{\mathcal{H}}^l = \{\bar{H}_i^l\}_{i=0}^{M-1}$ is the set of node embeddings $\bar{H}_i$ for $M$ different patches, $\text{Readout}$ is the readout function, and $D^H$ is the dimension of the output node embeddings. The readout function is defined to be a `MAX`, `MEAN`, or `LAST` pooling layer.

### 3.5 Improving Over-Smoothing by Alternating Architecture

Over-smoothing is a critical problem in graph representation learning, where MPNNs fall short in encoding long-range dependencies beyond a few layers of message-passing. This issue is magnified in dynamic graphs when temporal long-range dependencies intersect with structural patterns. MPNNs typically fall into the over-smoothing regime beyond a few layers (e.g., 3), which may not be sufficient to capture long-range temporal dynamics. We propose to address this problem by letting the Transformer widen up the temporal contextual node-wise scope beyond a few hops in an alternating manner. For instance, a 3-layer MPNN encoder can reach patterns up to 9 hops away in Todyformer.

Table 1: Future Link Prediction performance in AP (Mean ± Std) on the test set.

| Setting | Model | MOOC | LastFM | Enron | UCI | SocialEvol. |
|---|---|---|---|---|---|---|
| Transductive | JODIE | $0.797 \pm 0.01$ | $0.691 \pm 0.010$ | $0.785 \pm 0.020$ | $0.869 \pm 0.010$ | $0.847 \pm 0.014$ |
| | DyRep | $0.840 \pm 0.004$ | $0.683 \pm 0.033$ | $0.795 \pm 0.042$ | $0.524 \pm 0.076$ | $0.885 \pm 0.004$ |
| | TGAT | $0.793 \pm 0.006$ | $0.633 \pm 0.002$ | $0.637 \pm 0.002$ | $0.835 \pm 0.003$ | $0.631 \pm 0.001$ |
| | TGN | $0.911 \pm 0.010$ | $0.743 \pm 0.030$ | $0.866 \pm 0.006$ | $0.843 \pm 0.090$ | $0.966 \pm 0.001$ |
| | CaW | $0.940 \pm 0.014$ | $0.903 \pm 1e\text{-}4$ | $0.970 \pm 0.001$ | $0.939 \pm 0.008$ | $0.947 \pm 1e\text{-}4$ |
| | NAT | $0.874 \pm 0.004$ | $0.859 \pm 1e\text{-}4$ | $0.924 \pm 0.001$ | $0.944 \pm 0.002$ | $0.944 \pm 0.010$ |
| | GraphMixer | $0.835 \pm 0.001$ | $0.862 \pm 0.003$ | $0.824 \pm 0.001$ | $0.932 \pm 0.006$ | $0.935 \pm 3e\text{-}4$ |
| | Dygformer | $0.892 \pm 0.005$ | $0.901 \pm 0.003$ | $0.926 \pm 0.001$ | $0.959 \pm 0.001$ | $0.952 \pm 2e\text{-}4$ |
| | DyG2Vec | $\underline{0.980 \pm 0.002}$ | $\underline{0.960 \pm 1e\text{-}4}$ | $\underline{0.991 \pm 0.001}$ | $\underline{0.988 \pm 0.007}$ | $\underline{0.987 \pm 2e\text{-}4}$ |
| | **Todyformer** | $\mathbf{0.992 \pm 7e\text{-}4}$ | $\mathbf{0.976 \pm 3e\text{-}4}$ | $\mathbf{0.995 \pm 6e\text{-}4}$ | $\mathbf{0.994 \pm 4e\text{-}4}$ | $\mathbf{0.992 \pm 1e\text{-}4}$ |
| Inductive | JODIE | $0.707 \pm 0.029$ | $0.865 \pm 0.03$ | $0.747 \pm 0.041$ | $0.753 \pm 0.011$ | $0.791 \pm 0.031$ |
| | DyRep | $0.723 \pm 0.009$ | $0.869 \pm 0.015$ | $0.666 \pm 0.059$ | $0.437 \pm 0.021$ | $0.904 \pm 3e\text{-}4$ |
| | TGAT | $0.805 \pm 0.006$ | $0.644 \pm 0.002$ | $0.693 \pm 0.004$ | $0.820 \pm 0.005$ | $0.632 \pm 0.005$ |
| | TGN | $0.855 \pm 0.014$ | $0.789 \pm 0.050$ | $0.746 \pm 0.013$ | $0.791 \pm 0.057$ | $0.904 \pm 0.023$ |
| | CaW | $0.933 \pm 0.014$ | $0.890 \pm 0.001$ | $0.962 \pm 0.001$ | $0.931 \pm 0.002$ | $0.950 \pm 1e\text{-}4$ |
| | NAT | $0.832 \pm 1e\text{-}4$ | $0.878 \pm 0.003$ | $0.949 \pm 0.010$ | $0.926 \pm 0.010$ | $0.952 \pm 0.006$ |
| | GraphMixer | $0.814 \pm 0.002$ | $0.821 \pm 0.004$ | $0.758 \pm 0.004$ | $0.911 \pm 0.004$ | $0.918 \pm 6e\text{-}4$ |
| | Dygformer | $0.869 \pm 0.004$ | $0.942 \pm 9e\text{-}4$ | $0.897 \pm 0.003$ | $0.945 \pm 0.001$ | $0.931 \pm 4e\text{-}4$ |
| | DyG2Vec | $\underline{0.938 \pm 0.010}$ | $\underline{0.979 \pm 0.006}$ | $\underline{0.987 \pm 0.004}$ | $\underline{0.976 \pm 0.002}$ | $\underline{0.978 \pm 0.010}$ |
| | **Todyformer** | $\mathbf{0.948 \pm 0.009}$ | $\mathbf{0.981 \pm 0.005}$ | $\mathbf{0.989 \pm 8e\text{-}4}$ | $\mathbf{0.983 \pm 0.002}$ | $\mathbf{0.9821 \pm 0.005}$ |

## 4 Experimental Evaluation

In this section, we evaluate the generalization performance of Todyformer through a rigorous empirical assessment spanning a wide range of benchmark datasets across downstream tasks. First, the experimental setup is explained, and a comparison with the state-of-the-art (SoTA) on dynamic graphs is provided. Next, the quantitative results are presented. Later, in-depth comparative analysis and ablation studies are provided to further highlight the role of the design choices in this work.

### 4.1 Experimental Setup

**Baselines**: The performance of Todyformer is compared with a wide spectrum of dynamic graph encoders, ranging from random-walk based to attentive memory-based approaches: DyRep (Trivedi et al., 2019), JODIE (Kumar et al., 2019), TGAT (Xu et al., 2020), TGN (Rossi et al., 2020), CaW (Wang et al., 2021), NAT (Luo & Li, 2022), and DyG2Vec (Alomrani et al., 2022). CAW samples temporal random walks and learns temporal motifs by counting node occurrences in each walk. NAT constructs temporal node representations using a cache that stores a limited set of historical interactions for each node. DyG2Vec introduces a window-based MPNN that attentively aggregates messages in a window of recent interactions. Recently, GraphMixer (Cong et al., 2023) has been presented as a simple yet effective MLP-Mixer-based dynamic graph encoder. Dygformer (Yu et al., 2023) also presents a Transformer architecture that encodes the one-hop node neighborhoods. In the Dygformer framework, patchifying entails the application of historical interaction analysis to a specific node's context. In other words, the interaction history of a node within the one-hop context is retrieved and encoded using Transformers. In contrast to Dygformer's node-level approach, our methodology involves the extraction of subgraphs and subsequent higher-order message-passing within a patch, without strict emphasis on individual one-hop node neighborhoods. Consequently, our approach to patchifying and tokenization operates at the graph level where the input to Transformers is the encoded node embedding returned from the local encoding. This process ensures that the embeddings generated from MPNN are updated utilizing message-passing information from all nodes within the corresponding subgraphs.

**Downstream Tasks**: We evaluate all models on both FLP and DNC. In FLP, the goal is to predict the probability of future edges occurring given the source and destination nodes, and the timestamp. For each positive edge, we sample a negative edge that the model is trained to predict as negative. The DNC task

Table 2: Future Link Prediction performance on the test set of TGBL datasets measured using Mean Reciprocal Rank (MRR). The baseline results are directly taken from Huang et al. (2023).

| Model | Wiki | Review | Coin | Comment | Flight | Avg. Rank ↓ |
|---|---|---|---|---|---|---|
| Dyrep | $0.366 \pm 0.014$ | $0.367 \pm 0.013$ | $0.452 \pm 0.046$ | $0.289 \pm 0.033$ | $0.556 \pm 0.014$ | 4.4 |
| TGN | $0.721 \pm 0.004$ | $\mathbf{0.532 \pm 0.020}$ | $0.586 \pm 0.037$ | $0.379 \pm 0.021$ | $0.705 \pm 0.020$ | 2 |
| CAW | $\mathbf{0.791 \pm 0.015}$ | $0.194 \pm 0.004$ | $OOM$ | $OOM$ | $OOM$ | 4.4 |
| TCL | $0.712 \pm 0.007$ | $0.200 \pm 0.010$ | $OOM$ | $OOM$ | $OOM$ | 4.8 |
| GraphMixer | $0.701 \pm 0.014$ | $0.514 \pm 0.020$ | $OOM$ | $OOM$ | $OOM$ | 4.4 |
| EdgeBank | $0.641$ | $0.0836$ | $0.1494$ | $0.364$ | $0.580$ | 4.6 |
| **Todyformer** | $0.7738 \pm 0.004$ | $0.5104 \pm 86e\text{-}4$ | $\mathbf{0.689 \pm 18e\text{-}4}$ | $\mathbf{0.762 \pm 98e\text{-}4}$ | $\mathbf{0.777 \pm 0.014}$ | **1.6** |

involves predicting the ground-truth label of the source node of a future interaction. Both tasks are trained using the binary cross-entropy loss function. For FLP, we evaluate all models in both transductive and inductive settings. The latter is a more challenging setting where a model makes predictions on unseen nodes. The Average Precision (AP) and the Area Under the Curve (AUC) metrics are reported for the FLP and DNC tasks, respectively. DNC is evaluated using AUC due to the class imbalance issue.

**Datasets**: In the first set of experiments, we use 5 real-world datasets for FLP: MOOC, and LastFM (Kumar et al., 2019); SocialEvolution, Enron, and UCI (Wang et al., 2021). Three real-world datasets including Wikipedia, Reddit, MOOC (Kumar et al., 2019) are used for DNC as well. These datasets span a wide range of the number of nodes and interactions, timestamp ranges, and repetition ratios. The dataset statistics are presented in Appendix A.2. We employ the same 70%-15%-15% chronological split for all datasets, as outlined in Wang et al. (2021). The datasets are split differently under two settings: Transductive and Inductive. All benchmark datasets are publicly available. We follow similar experimental setups to Alomrani et al. (2022); Wang et al. (2021) on these datasets to split them into training, validation, and test sets under the transductive and inductive settings.

In the second set of experiments, we evaluate Todyformer on the Temporal Graph Benchmark for link prediction datasets (TGBL) (Huang et al., 2023). The goal is to target large-scale and real-world experimental setups with a higher number of negative samples generated based on two policies: random and historical. The deliberate inclusion of such negative edges aims to address the substantial bias inherent in negative sampling techniques, which can significantly affect model performance. Among the five datasets, three are extra-large-scale, where model training on a regular setup may take weeks of processing. We follow the experimental setups similar to Huang et al. (2023) to evaluate our model on TGBL (e.g., number of trials or negative sampling).

**Model Hyperparameters**: Todyformer has a large number of hyperparameters to investigate. There are common design choices, such as activation layers, normalization layers, and skip connections that we assumed the results are less sensitive to in order to reduce the total number of trials. We chose $L = 3$ for the number of blocks in the encoder. The GNN and Transformers have three and two layers, respectively. The neighbor sampler in the local encoder uniformly samples $(64, 1, 1)$ number of neighbors for 3 hops. The model employs uniform sampling within the window instead of selecting the latest $N$ neighbors of a node (Xu et al., 2020; Rossi et al., 2020). For the DNC task, following prior work by Rossi et al. (2020), the decoder is trained on top of the frozen encoder pre-trained on FLP.

## 4.2 Experimental Results

**Future Link Prediction**: We present a comparative analysis of AP scores on the test set for future link prediction (both transductive and inductive) across several baselines in Table 1. Notably, a substantial performance gap is evident in the transductive setting, with Todyformer outperforming the second-best model by margins exceeding 1.2%, 1.6%, 0.6%, and 0.5% on the MOOC, LastFM, UCI, and SocialEvolve datasets, respectively. Despite the large scale of the SocialEvolve dataset with around 2 million edges, our model achieves SoTA performance on this dataset. This observation reinforces the conclusions drawn in Xu et al. (2020), emphasizing the pivotal role played by recent temporal links in the future link prediction task. Within the inductive setting, Todyformer continues to exhibit superior performance across all datasets. The challenge posed by predicting links over unseen nodes impacts the overall performance of most methods. However, Todyformer consistently outperforms the baselines' results on all datasets in Table 1. These empirical results support the hypothesis that model expressiveness has significantly improved while enhancing the generalization under the two experimental settings. Additionally, Todyformer outperforms the two latest

Table 3: Dynamic Node Classification performance in AUC (Mean ± Std) on the test set. Avg. Rank reports the mean rank of a method across all datasets.

| Model | Wikipedia | Reddit | MOOC | Avg. Rank ↓ |
|---|---|---|---|---|
| TGAT | $0.800 \pm 0.010$ | $\mathbf{0.664 \pm 0.009}$ | $0.673 \pm 0.006$ | 3.6 |
| JODIE | $0.843 \pm 0.003$ | $0.566 \pm 0.016$ | $0.672 \pm 0.002$ | 4.6 |
| Dyrep | $\mathbf{0.873 \pm 0.002}$ | $0.633 \pm 0.008$ | $0.661 \pm 0.012$ | 4 |
| TGN | $0.828 \pm 0.004$ | $0.655 \pm 0.009$ | $0.674 \pm 0.007$ | 3.3 |
| DyG2Vec | $0.824 \pm 0.050$ | $0.649 \pm 0.020$ | $\mathbf{0.785 \pm 0.005}$ | 3.3 |
| **Todyformer** | $0.861 \pm 0.017$ | $0.656 \pm 0.005$ | $0.745 \pm 0.009$ | **2** |

SoTA methods, namely GraphMixer (Cong et al., 2023) and Dygformer (Yu et al., 2023). The results further validate that dynamic graphs require encoding of long-range dependencies that cannot be simply represented by short-range one-hop neighborhoods. This verifies that multi-scale encoders like Todyformer are capable of learning inductive biases across various domains.

Additionally, the performance of Todyformer on two small and three large TGBL datasets is presented in Table 2. On extra-large TGBL datasets (Coin, Comment, and Flight), Todyformer outperforms the SoTA with significant margins, exceeding 11%, 39%, and 7%, respectively. This interestingly supports the hypothesis behind the expressive power of the proposed model to scale up to the data domains with extensive long-range interactions. In the case of smaller datasets like TGBL-Wiki and TGBL-Review, our approach attains the second and third positions in the ranking, respectively. It should be noted that the hyperparameter search has not been exhausted during experimental evaluation. The average ranking reveals that Todyformer is ranked first, followed by TGN in the second place in this challenging experimental setup. Notably, datasets like TGBLFlight, TGBLCoin, and TGBLComment, characterized by imbalanced labels, represent particularly challenging tasks for models, making the achievements of todyformer even more remarkable. Additionally, Figure 7 depicts experiments conducted with a large number of layers. Notably, the figure illustrates that the alternating mode results in a slight performance increase as the number of layers grows. Conversely, stacking layers of MPNN-based models such as dyg2vec leads to a drop in model performance.

**Dynamic Node classification**: Todyformer has undergone extensive evaluation across three datasets dedicated to node classification. In these datasets, dynamic sparse labels are associated with nodes within a defined time horizon after interactions. This particular task grapples with a substantial imbalanced classification challenge. Table 3 presents the AUC metric, known for its robustness toward class imbalance, across various methods on the three datasets. Notably, Todyformer demonstrates remarkable performance, trailing the best by only 4% on the MOOC dataset and 1% on both the Reddit and Wikipedia datasets. Across all datasets, Todyformer consistently secures the second-best position. However, it is important to acknowledge that no model exhibits consistent improvement across all datasets, primarily due to the presence of data imbalance issues inherent in anomaly detection tasks (Ranshous et al., 2015). To establish the ultimate best model, we have computed the average ranks of various methods. Todyformer emerges as the top performer with an impressive rank of 2, validating the overall performance improvement.

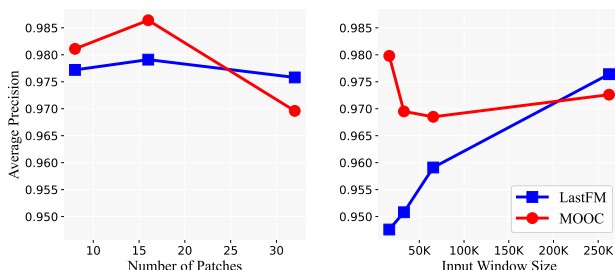

Figure 3: Sensitivity analysis on the number of patches and input window size values on MOOC and LastFM. The plot on the left has a fixed input window size of 262144, while the one on the right has 32 patches.

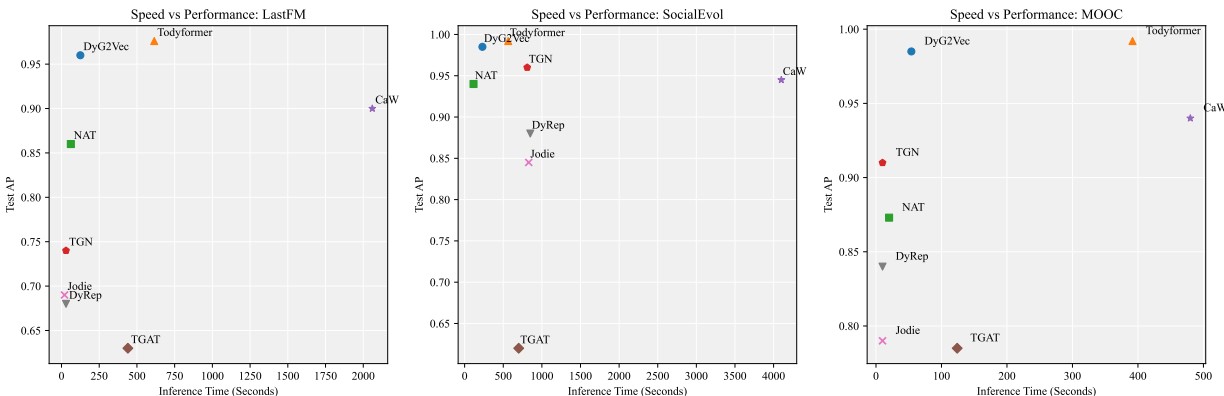

Figure 4: The performance versus inference time across LastFM, SocialEvol, and MOOC datasets.

## 4.3 Ablation Studies and sensitivity analysis

We conducted an evaluation of the model's performance across various parameters and datasets to assess the sensitivity of the major hyperparameters. Figure 3 illustrates the sensitivity analysis regarding the window size and the number of patches, with one parameter remaining constant while the other changes. As highlighted in Xu et al. (2020), recent and frequent interactions display enhanced predictability of future interactions. This predictability is particularly advantageous for datasets with extensive long-range dependencies, favoring the utilization of larger window size values to capture recurrent patterns. Conversely, in datasets where recent critical interactions reflect importance, excessive emphasis on irrelevant information becomes prominent when employing larger window sizes. Our model, complemented by uniform neighbor sampling, achieves a balanced equilibrium between these contrasting sides of the spectrum. As an example, the right plot in Figure 3 demonstrates that with a fixed number of patches (i.e., 32), an increase in window size leads to a corresponding increase in the validation AP metric on the LastFM dataset. This trend is particularly notable in LastFM, which exhibits pronounced long-range dependencies, in contrast to datasets like MOOC and UCI with medium- to short-range dependencies.

In contrast, in Figure 3 on the left side, with a window size of 262k, we vary the number of patches. Specifically, for the MOOC dataset, performance exhibits an upward trajectory with an increase in the number of patches from 8 to 16; however, it experiences a pronounced decline when the number of patches reaches 32. This observation aligns with the inherent nature of MOOC datasets, characterized by their relatively high density and reduced prevalence of long-range dependencies. Conversely, when considering LastFM data, the model maintains consistently high performance even with 32 patches. In essence, this phenomenon underscores the model's resilience on datasets featuring extensive long-range dependencies, illustrating a trade-off between encoding local and contextual features by tweaking the number of patches.

In Table 4, we conducted ablation studies on the major design choices of the encoding network to assess the roles of the three hyperparameters separately: a) Global encoder, b) Alternating mode, and c) Positional Encoding. Across the four datasets, the alternating approach exhibits significant performance variation compared to others, ensuring the mitigation of over-smoothing and the capturing of long-range dependencies. The outcomes of the single-layer vanilla transformer as a global encoder attain the second-best position, affirming the efficacy of our global encoder in enhancing expressiveness. Finally, the global encoder without PE closely resembles the model with only a local encoder (i.e., DyG2Vec MPNN model).

## 4.4 Training/Inference Speed

In this section, an analysis of Figure 4 is provided, depicting the performance versus inference time across three sizable datasets. Considering the delicate trade-off between performance and complexity, our model surpasses all others in terms of Average Precision (AP) while concurrently positioning in the left segment

Table 4: Ablation studies on three major components: Global Encoder (G. E.), Positional Encoder (P. E.), and number of alternating blocks (Alt. 3).

| Dataset | G. E. | P. E. | Alt. 3 | AP |
|---|---|---|---|---|
| MOOC | ✗ | ✗ | ✗ | 0.980 |
|  | ✓ | ✗ | ✗ | 0.981 |
|  | ✓ | ✓ | ✗ | 0.987 |
|  | ✓ | ✓ | ✓ | **0.992** |
| LastFM | ✗ | ✗ | ✗ | 0.960 |
|  | ✓ | ✗ | ✗ | 0.961 |
|  | ✓ | ✓ | ✗ | 0.965 |
|  | ✓ | ✓ | ✓ | **0.976** |
| UCI | ✗ | ✗ | ✗ | 0.981 |
|  | ✓ | ✗ | ✗ | 0.983 |
|  | ✓ | ✓ | ✗ | 0.987 |
|  | ✓ | ✓ | ✓ | **0.993** |
| SocialEvolution | ✗ | ✗ | ✗ | 0.987 |
|  | ✓ | ✗ | ✗ | 0.987 |
|  | ✓ | ✓ | ✗ | 0.989 |
|  | ✓ | ✓ | ✓ | **0.991** |

of the diagrams, denoting the lowest inference time. Notably, as depicted in Figure 4, Todyformer remains lighter and less complex than state-of-the-art (SOTA) models like CAW across all datasets.

## 5 Conclusion

We propose Todyformer, a tokenized graph Transformer for dynamic graphs, where over-smoothing and over-squashing are empirically improved through a local and global encoding architecture. We present how to adapt the best practices of Transformers in various data domains (e.g., Computer Vision) to dynamic graphs in a principled manner. The primary novel components are patch generation, structure-aware tokenization using typical MPNNs that locally encode neighborhoods, and the utilization of Transformers to aggregate global context in an alternating fashion. The consistent experimental gains across different experimental settings empirically support the hypothesis that the SoTA dynamic graph encoders severely suffer from over-squashing and over-smoothing phenomena, especially on the real-world large-scale datasets introduced in TGBL. We hope Todyformer sheds light on the underlying aspects of dynamic graphs and opens up the door for further principled investigations on dynamic graph transformers.

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

# A Appendix

## A.1 Table of Notations

Table 5: Overall notation table of the main symbols in the paper.

| basic notations | |
| --- | --- |
| $N$ | The number of nodes. |
| $E$ | The number of edges. |
| $D^H$ | The dimension of the embedding space. |
| $W$ | The window size. |
| $L$ | The number of blocks. |
| **Sets and Matrices** | |
| $\mathcal{G}$ | A Continuous-Time Dynamic Graph (CTDG). |
| $\mathcal{E}$ | The sequence of interactions. |
| $\mathcal{V}$ | The set of vertices. |
| $\mathcal{X}^E$ | The node features. |
| $\mathcal{X}^v$ | The edge features. |
| $\mathcal{P}$ | The set of patches. |
| $\mathcal{X}$ | The set of node features for all patches. |
| $Q$ | The query. |
| $K$ | The key. |
| $V$ | The value. |
| $\boldsymbol{H}$ | The final node embeddings. |
| $\mathcal{H}^l$ | The set of node embeddings after applying the local encoder. |
| $\bar{H}^l$ | The set of node embeddings after applying the global encoder. |
| $H^l$ | The set of node embeddings after packing. |
| $\boldsymbol{Z}$ | The predictions for the ground truth labels. |
| $\bar{\mathcal{H}}^l$ | The set of node embeddings after unpacking. |
| **Learnable Parameters** | |
| $\theta$ | The weight parameters of the encoder. |
| $\gamma$ | The weight parameters of the decoder. |
| $W_q$ | The learnable matrix of the query. |
| $W_k$ | The learnable matrix of the key. |
| $W_v$ | The learnable matrix of the value. |
| **Functions and Modules** | |
| $\text{ENCODER}_\theta$ | The dynamic graph encoder. |
| $\text{DECODER}_\gamma$ | The decoder. |
| LOCALENCODER | The set of local encoding modules. |
| GLOBALENCODER | The set of global encoding modules. |
| PATCHIFIER | The graph partitioning module. |
| $\text{TOKENIZER}^l$ | The tokenization module. |
| $\text{PACKER}^l$ | The packing module. |
| $\text{UNPACKER}^l$ | The unpacking module. |
| $\text{POSITIONALENCODER}^l$ | The positional encoder. |
| $\text{TRANS}^l$ | The Transformer module. |
| READOUT | The readout function. |

## A.2 Dataset Statistics

In this section, we provide an overview of the statistics pertaining to two distinct sets of datasets utilized for the tasks of Future Link Prediction (FLP) and Dynamic Node Classification (DNC). The initial set, detailed in Table 6, presents information regarding the number of nodes, edges, and unique edges across seven datasets featured in Table 1 and Table 3. For these three datasets, namely Reddit, Wikipedia, and MOOC, all edge features have been incorporated, where applicable. Furthermore, within this table, the last column represents the percentage of Repetitive Edges, which signifies the proportion of edges that occur more than once within the dynamic graph.

Table 6: Dynamic Graph Datasets. **% Repetitive Edges**: % of edges which appear more than once in the dynamic graph.

| Dataset | # Nodes | # Edges | # Unique Edges | Edge Features | Node Labels | Bipartite | % Repetitive Edges |
| --- | --- | --- | --- | --- | --- | --- | --- |
| Reddit | 11,000 | 672,447 | 78,516 | ✓ | ✓ | ✓ | 54% |
| Wikipedia | 9,227 | 157,474 | 18,257 | ✓ | ✓ | ✓ | 48% |
| MOOC | 7,144 | 411,749 | 178,443 | ✓ | ✓ | ✓ | 53% |
| LastFM | 1980 | 1,293,103 | 154,993 | | | ✓ | 68% |
| UCI | 1899 | 59,835 | 13838 | | | ✓ | 62% |
| Enron | 184 | 125,235 | 2215 | | | | 92% |
| SocialEvolution | 74 | 2,099,519 | 2506 | | | | 97% |

### A.2.1 Temporal Graph Benchmark (TGB) dataset

In this section, we present the characteristics of datasets as proposed by the Dynamic Graph Encoder Leaderboard (Huang et al., 2023). Similar to previous benchmark datasets, we have conducted comparisons regarding the number of nodes, edges, and types of graphs. Additionally, we report the Number of Steps

and the Surprise Index, as defined in Poursafaei et al. (2022), which illustrates the ratio of test edges that were not observed during the training phase.

Table 7: Statistics of TGBL Dynamic Graph Datasets

| Dataset | # Nodes | # Edges | # Steps | Edge Features | Bipartite | Surprise Index Poursafaei et al. (2022) |
|---------|---------|---------|---------|---------------|-----------|------------------------------------------|
| Wiki | 9,227 | 157,474 | 152,757 | ✓ | ✓ | 0.108 |
| Review | 352,637 | 4,873,540 | 6,865 | ✓ | ✓ | 0.987 |
| Coin | 638,486 | 22,809,486 | 1,295,720 | ✓ | | 0.120 |
| Comment | 994,790 | 44,314,507 | 30,998,030 | ✓ | | 0.823 |
| Flight | 18143 | 67,169,570 | 1,385 | ✓ | | 0.024 |

## A.3  Implementation details

In this section, we elucidate the intricacies of our implementation, providing a comprehensive overview of the specific parameters our model accommodates during hyperparameter optimization. Subsequently, we delve into a discussion of the optimal configurations and setups that yield the best performance for our proposed architecture.

Furthermore, in addition to an in-depth discussion of the baselines incorporated into our paper, we also offer a comprehensive overview of the respective hyperparameter configurations in this section. We are confident that with the open-sourcing of our code upon acceptance and the thorough descriptions of our model and baseline methodologies presented in the paper, our work is fully reproducible.

### A.3.1  Supplementary Methodology

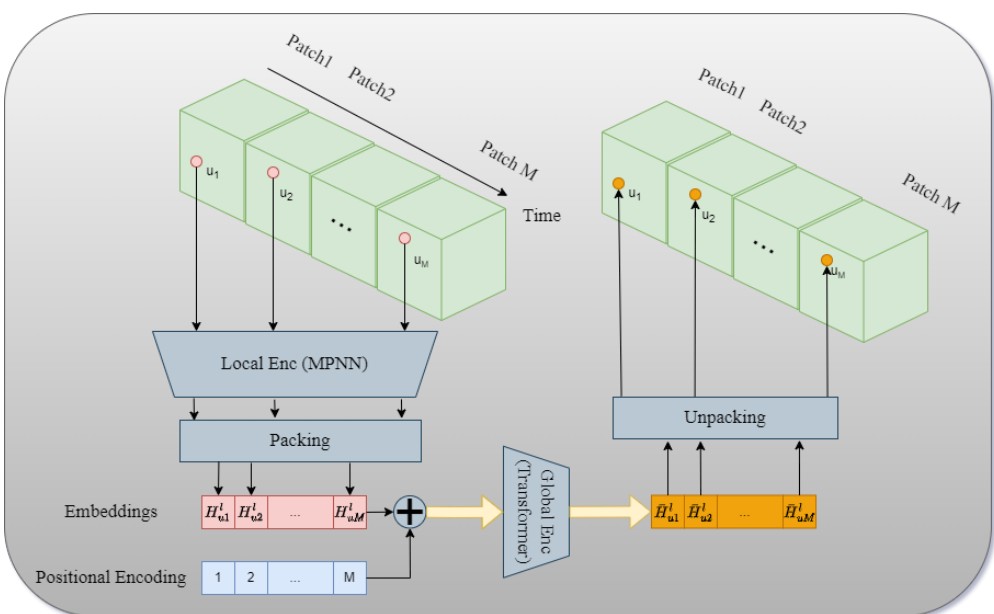

Figure 5: In-depth illustration of model architecture, encompassing packing and unpacking mechanisms, positional encoding, and encoder components.

### A.3.2 Evaluation Protocol

**Transductive Setup:** Under the transductive setting, a dataset is split normally by time, i.e., the model is trained on the first 70% of links, validated on %15 and tested on the rest.

**Inductive Setup:** In the inductive setting, we strive to test the model's prediction performance on edges with unseen nodes. Therefore, following (Wang et al., 2021), we randomly assign 10% of the nodes to the validation and test sets and remove any interactions involving them in the training set. Additionally, to ensure an inductive setting, we remove any interactions not involving these nodes from the test set.

### A.3.3 Loss Function

As previously discussed in the main body of this paper, we focus on two specific downstream tasks: FLP, and DNC. For the former, we employ the binary cross-entropy loss, while for the latter, our model is trained through the minimization of the cross-entropy loss function. The formula for the binary cross-entropy loss is presented below:

$$H(y, \hat{y}) = - (y \cdot \log(\hat{y}) + (1 - y) \cdot \log(1 - \hat{y})) \tag{8}$$

where $y \in \{0, 1\}$ is the true label, and $\hat{y}$ is the predicted probability that the instance belongs to class 1. Moreover, the formulation of the cross-entropy loss is as follows:

$$H(y, \hat{y}) = - \sum_i y_i \cdot \log(\hat{y}_i) \tag{9}$$

where $i$ represents the index over all classes, $y_i$ is the true probability of the sample belonging to class $i$, encoded as a one-hot vector. It is 1 for the true class and 0 for all other classes. Finally, $\hat{y}_i$ is the predicted probability of the sample belonging to class $i$.

### A.3.4 Best Hyperparameters for Benchmark datasets.

Table 8 displays the hyperparameters that have been subjected to experimentation and tuning for each dataset. For each parameter, a range of values has been tested as follows:

- Window Size (W): This parameter signifies the window length chosen for selecting the input subgraph based on edge timestamps. It falls within the range of $\in \{$ 16384, 32768 ,65536, 262144 $\}$.

- Number of Patches: This parameter indicates the count of equal and non-overlapping chunks for each input subgraph. It is the range of $\in \{8, 16, 32\}$.

- #Local Encoders: This parameter represents the number of local encoder layers within each block, and its value falls within the range of $\in \{1, 2\}$.

- Neighbor Sampling (NS) mode: $\in \{uniform, last\}$. In the case of a uniform Neighbor Sampler (NS), it uniformly selects samples from the 1-hop interactions of a given node. Conversely, in last mode, it samples from the most recent interactions.

- Anchor Node Mode: $\in \{GlobalTarget, LocalInput, LocalTarget\}$ depending on the mechanism of neighbor sampling we can sample from nodes within all patches (LocalInput), nodes within the next patch (LocalTarget), or global target nodes (GlobalTarget).

- Batch Size: $\in \{8, 16, 32, 64\}$

- Positional Encoding: $\in \{SineCosine, Time2Vec, Identity, Linear\}$

SineCosine is utilized as the Positional Encoding (PE) method following the experiments conducted in Appendix A.4.3.

**Selecting Best Checkpoint:** Throughout all experiments, the models undergo training for a duration of 100 epochs, with the best checkpoints selected for testing based on their validation Average Precision (AP) performance.

Table 8: Best parameters of the model pipeline after hyperparameter search.

| Dataset | Window Size ($W$) | Number of Patches | #Local Encoders | NS Mode | Anchor Node Mode | Batch Size |
|---|---|---|---|---|---|---|
| Reddit | 262144 | 32 | 2 | uniform | GlobalTarget | 200 |
| Wikipedia | 65536 | 8 | 2 | uniform | GlobalTarget | 200 |
| MOOC | 65536 | 8 | 2 | uniform | GlobalTarget | 200 |
| LastFM | 262144 | 32 | 2 | uniform | GlobalTarget | 200 |
| UCI | 65536 | 8 | 2 | uniform | GlobalTarget | 200 |
| Enron | 65536 | 8 | 2 | uniform | GlobalTarget | 200 |
| SocialEvolution | 65536 | 8 | 2 | uniform | GlobalTarget | 200 |

### A.3.5 Best Hyperparameters for TGB dataset

In this section, we present the optimal hyperparameters used in our architecture design for each TGBL dataset. We conducted hyperparameter tuning for all TGBL datasets; however, due to time constraints, we explored a more limited set of parameters for the large-scale dataset. Despite Todyformer outperforming its counterparts on these datasets, there remains potential for further improvement through an extensive hyperparameter search.

Table 9: Optimal window size $W$ for downstream training.

| Dataset | Window Size ($W$) | Number of Patches | First-hop NS size | NS Mode | Anchor Node Mode | Batch Size |
|---|---|---|---|---|---|---|
| Wiki | 262144 | 32 | 256 | uniform | GlobalTarget | 32 |
| Review | 262144 | 32 | 64 | uniform | GlobalTarget | 64 |
| Comment | 65536 | 8 | 64 | uniform | GlobalTarget | 256 |
| Coin | 65536 | 8 | 64 | uniform | GlobalTarget | 96 |
| Flight | 65536 | 8 | 64 | uniform | GlobalTarget | 128 |

### A.4 More Experimental Result

In this section, we present the additional experiments conducted and provide an analysis of the derived results and conclusions.

### A.4.1 FLP result on Benchmark Datasets

Table 10 is an extension of Table 1, now incorporating the Wikipedia and Reddit datasets. Notably, for these two datasets, Todyformer attains the highest test Average Precision (AP) score in the Transductive setup. However, it secures the second-best and third-best positions in the Inductive setup for these Wikipedia and Reddit respectively. While the model does not attain the top position on these two datasets for inductive setup, its performance is only marginally below that of state-of-the-art (SOTA) models, which have previously achieved accuracy levels exceeding 99% Average Precision (AP).

### A.4.2 FLP validation result on TGBL dataset

As discussed in the paper, Todyformer has been compared to baseline methods using the TGBL dataset. Table 11 represents an extension of Table 2 specifically for validation (MRR). The results presented in both tables are in line with counterpart methods outlined by Huang et al. (2023). It is important to note that for the larger datasets, TCL, GraphMIxer, and EdgeBank were found to be impractical due to memory constraints, as mentioned in the paper.

### A.4.3 Complementary Sensitivity Analysis and Ablation Study

In this section, we have presented the specifics of sensitivity and ablation experiments, which, while of lesser significance in our hyper-tuning mechanism, contribute valuable insights. In all tables, the Average Precision scores reported in the table are extracted from the same epoch on the validation set. Table 12 showcases

Table 10: Future Link Prediction performance in AP (Mean ± Std). **Bold** font and ul font represent first-best and second-best performance respectively.

| Setting | Model | Wikipedia | Reddit | MOOC | LastFM | Enron | UCI | SocialEvol. |
|---|---|---|---|---|---|---|---|---|
| Transductive | JODIE | $0.956 \pm 0.002$ | $0.979 \pm 0.001$ | $0.797 \pm 0.01$ | $0.691 \pm 0.010$ | $0.785 \pm 0.020$ | $0.869 \pm 0.010$ | $0.847 \pm 0.014$ |
| | DyRep | $0.955 \pm 0.004$ | $0.981 \pm 1e\text{-}4$ | $0.840 \pm 0.004$ | $0.683 \pm 0.033$ | $0.795 \pm 0.042$ | $0.524 \pm 0.076$ | $0.885 \pm 0.004$ |
| | TGAT | $0.968 \pm 0.001$ | $0.986 \pm 3e\text{-}4$ | $0.793 \pm 0.006$ | $0.633 \pm 0.002$ | $0.637 \pm 0.002$ | $0.835 \pm 0.003$ | $0.631 \pm 0.001$ |
| | TGN | $0.986 \pm 0.001$ | $0.985 \pm 0.001$ | $0.911 \pm 0.010$ | $0.743 \pm 0.030$ | $0.866 \pm 0.006$ | $0.843 \pm 0.090$ | $0.966 \pm 0.001$ |
| | CaW | $0.976 \pm 0.007$ | $0.988 \pm 2e\text{-}4$ | $0.940 \pm 0.014$ | $0.903 \pm 1e\text{-}4$ | $0.970 \pm 0.001$ | $0.939 \pm 0.008$ | $0.947 \pm 1e\text{-}4$ |
| | NAT | $0.987 \pm 0.001$ | $0.991 \pm 0.001$ | $0.874 \pm 0.004$ | $0.859 \pm 1e\text{-}4$ | $0.924 \pm 0.001$ | $0.944 \pm 0.002$ | $0.944 \pm 0.010$ |
| | GraphMixer | $0.974 \pm 0.001$ | $0.975 \pm 0.001$ | $0.835 \pm 0.001$ | $0.862 \pm 0.003$ | $0.824 \pm 0.001$ | $0.932 \pm 0.006$ | $0.935 \pm 3e\text{-}4$ |
| | Dygformer | $0.991 \pm 0.0001$ | $0.992 \pm 0.0001$ | $0.892 \pm 0.005$ | $0.901 \pm 0.003$ | $0.926 \pm 0.001$ | $0.959 \pm 0.001$ | $0.952 \pm 2e\text{-}4$ |
| | DyG2Vec | $\underline{0.995 \pm 0.003}$ | $\underline{0.996 \pm 2e\text{-}4}$ | $\underline{0.980 \pm 0.002}$ | $\underline{0.960 \pm 1e\text{-}4}$ | $\underline{0.991 \pm 0.001}$ | $\underline{0.988 \pm 0.007}$ | $\underline{0.987 \pm 2e\text{-}4}$ |
| | **Todyformer** | $\mathbf{0.996 \pm 2e\text{-}4}$ | $\mathbf{0.998 \pm 8e\text{-}5}$ | $\mathbf{0.992 \pm 7e\text{-}4}$ | $\mathbf{0.976 \pm 3e\text{-}4}$ | $\mathbf{0.995 \pm 6e\text{-}4}$ | $\mathbf{0.994 \pm 4e\text{-}4}$ | $\mathbf{0.992 \pm 1e\text{-}4}$ |
| Inductive | JODIE | $0.891 \pm 0.014$ | $0.865 \pm 0.021$ | $0.707 \pm 0.029$ | $0.865 \pm 0.03$ | $0.747 \pm 0.041$ | $0.753 \pm 0.011$ | $0.791 \pm 0.031$ |
| | DyRep | $0.890 \pm 0.002$ | $0.921 \pm 0.003$ | $0.723 \pm 0.009$ | $0.869 \pm 0.015$ | $0.666 \pm 0.059$ | $0.437 \pm 0.021$ | $0.904 \pm 3e\text{-}4$ |
| | TGAT | $0.954 \pm 0.001$ | $0.979 \pm 0.001$ | $0.805 \pm 0.006$ | $0.644 \pm 0.002$ | $0.693 \pm 0.004$ | $0.820 \pm 0.005$ | $0.632 \pm 0.005$ |
| | TGN | $0.974 \pm 0.001$ | $0.954 \pm 0.002$ | $0.855 \pm 0.014$ | $0.789 \pm 0.050$ | $0.746 \pm 0.013$ | $0.791 \pm 0.057$ | $0.904 \pm 0.023$ |
| | CaW | $0.977 \pm 0.006$ | $0.984 \pm 2e\text{-}4$ | $0.933 \pm 0.014$ | $0.890 \pm 0.001$ | $0.962 \pm 0.001$ | $0.931 \pm 0.002$ | $0.950 \pm 1e\text{-}4$ |
| | NAT | $0.986 \pm 0.001$ | $0.986 \pm 0.002$ | $0.832 \pm 1e\text{-}4$ | $0.878 \pm 0.003$ | $0.949 \pm 0.010$ | $0.926 \pm 0.010$ | $0.952 \pm 0.006$ |
| | GraphMixer | $0.966 \pm 2e\text{-}4$ | $0.952 \pm 2e\text{-}4$ | $0.814 \pm 0.002$ | $0.821 \pm 0.004$ | $0.758 \pm 0.004$ | $0.911 \pm 0.004$ | $0.918 \pm 6e\text{-}4$ |
| | Dygformer | $0.985 \pm 3e\text{-}4$ | $\underline{0.988 \pm 2e\text{-}4}$ | $0.869 \pm 0.004$ | $0.942 \pm 9e\text{-}4$ | $0.897 \pm 0.003$ | $0.945 \pm 0.001$ | $0.931 \pm 4e\text{-}4$ |
| | DyG2Vec | $\mathbf{0.992 \pm 0.001}$ | $\mathbf{0.991 \pm 0.002}$ | $\underline{0.938 \pm 0.010}$ | $\underline{0.979 \pm 0.006}$ | $\underline{0.987 \pm 0.004}$ | $\underline{0.976 \pm 0.002}$ | $\underline{0.978 \pm 0.010}$ |
| | **Todyformer** | $\underline{0.989 \pm 6e\text{-}4}$ | $0.983 \pm 0.002$ | $\mathbf{0.948 \pm 0.009}$ | $\mathbf{0.981 \pm 0.005}$ | $\mathbf{0.989 \pm 8e\text{-}4}$ | $\mathbf{0.983 \pm 0.002}$ | $\mathbf{0.9821 \pm 0.005}$ |

the influence of varying input window sizes and patch sizes on two datasets. Table 13 illustrates the effects of various PEs, including SineCosine, Time2Vec (Kazemi et al., 2019), Identity, Linear, and a configuration utilizing Local Input as the Anchor Node Mode. The table presents a comparison of results for these different PEs. Notably, the architecture appears to be relatively insensitive to the type of PE used, as the results all fall within a similar range. However, it is worth mentioning that SineCosine PE slightly outperforms the others. Consequently, SineCosine PE will be selected as the primary module for all subsequent experiments.

In Table14, an additional ablation study has been conducted to elucidate the influence of positions tagged to each node before being input to the Positional Encoder module. Various mechanisms for adding positions are delineated as follows:

- Without PE: No position is utilized or tagged to the nodes.

- Random Index: An index is randomly generated and added to the embeddings of a given node.

- Patch Index: The index of the patch from which the embedding of the given node originates is used as a position.

- Edge Time: The most recent edge time within its patch is employed as a position.

- Edge Index: The index of the most recent interaction within the corresponding patch is utilized as a position.

As evident from the findings in this table, the validation performance exhibits high sensitivity to the positional encoder's outcomes. Specifically, the model without positional encoder (PE) and the model with random

Table 11: (Validation) Future Link Prediction performance in Validation MRR on TGB Leaderboard datasets.

| Model | TGBWiki | TGBReview | TGBCoin | TGBComment | TGBFlight | Avg. Rank ↓ |
|---|---|---|---|---|---|---|
| Dyrep | $0.411 \pm 0.015$ | $0.356 \pm 0.016$ | $0.512 \pm 0.014$ | $0.291 \pm 0.028$ | $0.573 \pm 0.013$ | 4.2 |
| TGN | $0.737 \pm 0.004$ | $\mathbf{0.465 \pm 0.010}$ | $\underline{0.607 \pm 0.014}$ | $0.356 \pm 0.019$ | $\underline{0.731 \pm 0.010}$ | 2.2 |
| CAWN | $\underline{0.794 \pm 0.014}$ | $0.201 \pm 0.002$ | $OOM$ | $OOM$ | $OOM$ | 3 |
| TCL | $0.734 \pm 0.007$ | $0.194 \pm 0.012$ | $OOM$ | $OOM$ | $OOM$ | 5 |
| GraphMixer | $0.707 \pm 0.014$ | $0.411 \pm 0.025$ | $OOM$ | $OOM$ | $OOM$ | 4 |
| EdgeBank | $0.641$ | $0.0894$ | $0.1244$ | $\underline{0.388}$ | $0.492$ | 4.6 |
| **Todyformer** | $\mathbf{0.799 \pm 0.0092}$ | $\underline{0.4321 \pm 0.0040}$ | $\mathbf{0.6852 \pm 0.0021}$ | $\mathbf{0.7402 \pm 0.0037}$ | $\mathbf{0.7932 \pm 0.014}$ | **1.2** |

indices manifest the lowest performance among all available options. Consistent with our expectations from previous experiments, the patch index yields the highest performance, providing a compelling rationale for its incorporation into the architecture.

In Table 4, which presents the ablation study, we meticulously assess the impact of various components of the model architecture, particularly those associated with the alternating mode, such as packing and unpacking. This section has been comprehensively updated in the paper's appendix. The table delineates different scenarios:

- If the model lacks any of the three components (G.E, P.E, Alt.3), it signifies that the model solely relies on an MPNN-based encoder (local encoder).

- If it includes only G.E, it indicates the presence of packing alongside one block of Local and Global encoders without any unpacking.

- If G.E and P.E are present without Alt.3, this suggests the inclusion of one global and local encoder block coupled with patch positional encoding.

- Finally, if all components are present, it signifies the incorporation of three blocks of local and global encoders with positional encoding, as well as packing and unpacking modules within each block.

Table 12: Sensitivity analysis on the number of patches and the target window size.

| dataset | Input Window size | Number of Patches | Average Precision ↑ |
|---------|-------------------|-------------------|---------------------|
| LastFM  | 262144            | 8                 | 0.9772              |
| LastFM  | 262144            | 16                | 0.9791              |
| LastFM  | 262144            | 32                | 0.9758              |
| MOOC    | 262144            | 8                 | 0.9811              |
| MOOC    | 262144            | 16                | 0.9864              |
| MOOC    | 262144            | 32                | 0.9696              |
| LastFM  | 16384             | 32                | 0.9476              |
| LastFM  | 32768             | 32                | 0.9508              |
| LastFM  | 65536             | 32                | 0.9591              |
| LastFM  | 262144            | 32                | 0.9764              |
| MOOC    | 16384             | 32                | 0.9798              |
| MOOC    | 32768             | 32                | 0.9695              |
| MOOC    | 65536             | 32                | 0.9685              |
| MOOC    | 262144            | 32                | 0.9726              |

Table 13: Ablation study on positional encoding options on the MOOC Dataset. This table compares the validation performance at the same epoch across various setups.

| Positional Encoding | Anchor_Node_Mode | Average Precision ↑ |
|---------------------|------------------|---------------------|
| SineCosinePos       | global target    | 0.9901              |
| Time2VecPos         | global target    | 0.989               |
| IdentityPos         | global target    | 0.99                |
| LinearPos           | global target    | 0.9886              |
| SineCosinePos       | local input      | 0.9448              |

Table 14: Ablation study on the input of positional encoding on the MOOC Dataset. This table compares the validation performance at the same epoch across various types of positions tagged to nodes before PE layer.

| Positional Encoding (PE) Input | Average Precision ↑ |
|---|---|
| without PE | 0.9872 |
| random index | 0.9873 |
| patch index | 0.9889 |
| edge time | 0.9886 |
| edge index | 0.9877 |

## A.5 Computational Complexity

### A.5.1 Qualitative Analysis for Time and Memory Complexities

In this section, we delve into the detailed measurement and discussion of the computational complexity of Todyformer. Initially, we adopt the assumption that the time complexity of Transformers is $O(X^2)$ for an input sequence of length $X$. The primary complexity of Todyformer encompasses both the complexity of the Message Passing Neural Network (MPNN) component and the complexity of the Transformer. To elaborate further, assuming we have a sparse dynamic graph with temporal attributes, we can replace the complexity of MPNNs with $O(l \times (N+E))$, where N and E represent the number of nodes and edges within the temporal input subgraph, and $l$ is the number of MPNN layers for the Graph Neural Network (GNN) local encoder. In the transformer part, $N$ unique nodes are fed into the Multihead-Attention module. If the maximum length of the sequence fed to the Transformer is $N_a$, then the complexity of the Multihead-Attention module is $O(N_a^2)$. Notably, $N_a$ is at most equal to $M$, the number of patches. This scenario occurs when a node appears in all M patches and has interactions in all patches. Consequently, if $L$ is the number of blocks, the final complexity is given by:

$$O(L \times l \times (N+E) + L \times N \times M^2) \approx O(N+E) \tag{10}$$

The LHS part of Equation 10 can be simplified to RHS if we assume that $L$, $l$, and $M^2$ are negligible compared to $N$ and $E$. The RHS of this equation is the time complexity of GNNs for sparse graphs.

### A.5.2 Training/Inference Speed

In this section, an analysis of Figure 4 is provided, depicting the performance versus inference time across three sizable datasets. Considering the delicate trade-off between performance and complexity, our model surpasses all others in terms of Average Precision (AP) while concurrently positioning in the left segment of the diagrams, denoting the lowest inference time. Notably, as depicted in Figure 4, Todyformer remains lighter and less complex than state-of-the-art (SOTA) models like CAW across all datasets.

### A.6 Discussion on Over-Smoothing and Over-Squashing

In Figure 6, the blue curve illustrates the Average Precision performance of dynamic graph Message Passing Neural Networks (MPNNs) across varying numbers of layers. Notably, an observed trend indicates that as the number of layers increases, the performance experiences a decline—a characteristic manifestation of oversmoothing and oversquashing phenomena.

Within the same figure, the red circle dots represent the performance of MPNNs augmented with transformers, specifically Todyformer with a single block. It is noteworthy that the increase in the number of MPNN layers from 3 to 9 in this configuration results in a comparatively minor performance drop compared to traditional MPNNs.

Furthermore, the green stars denote the performance of Todyformer with an alternating mode, where the total number of MPNNs is 9, and three blocks are incorporated. In this setup, a transformer is introduced

after every 3 MPNN layers. Strikingly, this configuration outperforms all others, especially those that stack a similar number of MPNN layers without the insertion of a transformer layer in the middle of the architecture. This empirical observation serves as a significant study, highlighting the efficacy of our architecture in addressing oversmoothing and oversquashing challenges.

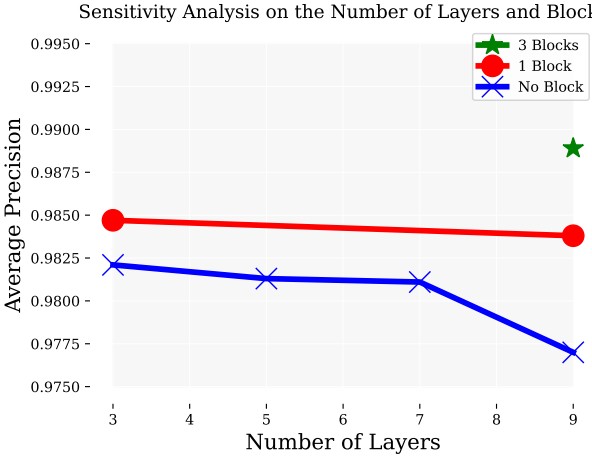

Figure 6: Sensitivity analysis on the number of layers and blocks conducted on the MOOC dataset.

To demonstrate the resilience of our model against oversmoothing, we have devised a novel experiment under a distinct setup. Due to constraints in computing resources and to accommodate models with a considerable number of layers on GPUs, we have opted for a smaller dataset. Furthermore, considering the significant stacking of layers (approximately 50), we have adjusted the neighbor sampling size to one for layers beyond the initial layer, primarily to mitigate memory consumption and ensure compatibility with GPU memory constraints.

In Figure 7, we present a comparative analysis of Average Precision (AP) scores across different models, including MPNN-based model such as dyg2vec, and todyformer, varying in the number of layers. Notably, the performance of MPNN exhibits a decline with an increase in the number of layers. Despite employing an approximation technique during neighbor sampling to accommodate a larger number of layers, the decline in performance is relatively modest. However, the observed decreasing trend suggests a hypothetical association with the issue of oversmoothing. Conversely, for todyformer, we observe a reverse pattern, with a minor improvement in performance as the number of layers increases. This empirical observation further strengthens the argument that todyformer is resilient to the effects of oversmoothing.

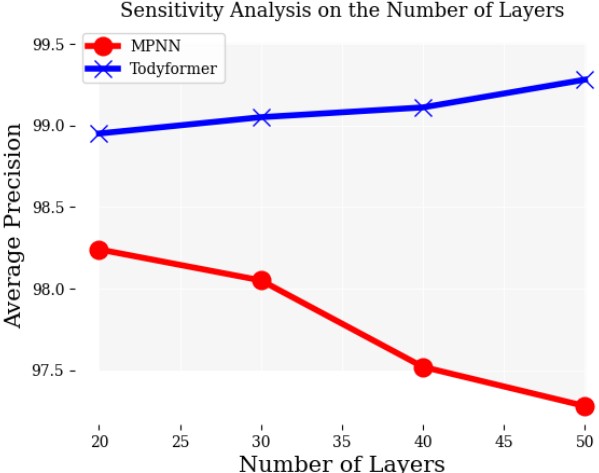

Figure 7: Sensitivity analysis on the number of layers conducted on the Enron dataset. The neighbor sampling size is configured to be one for subsequent layers following the initial layer.

