# OpenReview forum: "Todyformer: Towards Holistic Dynamic Graph Transformers with Structure-Aware Tokenization"
_TMLR — Accepted by TMLR_

### Review · Reviewer_5WPT · 2024-03-05

**Summary Of Contributions:**

The PDF paper introduces Todyformer, a novel Transformer-based neural network tailored for dynamic graphs. The paper addresses common issues in Temporal Graph Neural Networks, such as over-squashing and over-smoothing, by combining the local encoding capacity of Message-Passing Neural Networks with the global encoding of Transformers. Todyformer introduces innovative strategies like a patchifying paradigm for dynamic graphs, structure-aware parametric tokenization, temporal positional encoding in Transformers, and an encoding architecture that balances local and global contextualization to mitigate over-smoothing.
Experimental evaluations on public benchmark datasets demonstrate that Todyformer consistently outperforms state-of-the-art methods in downstream tasks, showcasing its effectiveness in capturing extensive temporal dependencies in dynamic graphs. The paper provides valuable insights into the design and performance of Todyformer, highlighting its potential to advance the field of dynamic graph analysis. However, important references are missing, raising novelty issue without discussion these references.

**Audience:**

Yes

**Claims And Evidence:**

Yes

**Requested Changes:**

1. While Todyformer outperforms the two latest state-of-the-art methods, GraphMixer and Dygformer, the paper does not provide a detailed comparison with a broader range of existing models, potentially limiting the scope of the evaluation. A thorough literature review is required for a high quality paper. For example, the following two papers are highly relevant and using similar strategies, thus should be discussed and compared:
Transformer-Style Relational Reasoning with Dynamic Memory Updating for Temporal Network Modeling. AAAI'21.
Spatio-Temporal Attentive RNN for Node Classification in Temporal Attributed Graphs. IJCAI'19.
Besides, the following paper is also relevant, should be discussed:
Adaptive Neural Network for Node Classification in Dynamic Networks. ICDM'19.
Refer to the "3.5 Improving Over-Smoothing by Alternating Architectur". The following paper is highly relevant, thus should be discussed:
Learning to Drop: Robust Graph Neural Network via Topological Denoising. WSDM'21.

2.  The paper acknowledges that no model exhibits consistent improvement across all datasets due to data imbalance issues inherent in anomaly detection tasks, suggesting that Todyformer may face challenges in handling imbalanced datasets. Thus, it is highly suggested to include an experiment to test the performance on imbalance dataset.

**Strengths And Weaknesses:**

Pros of Todyformer:
1: Todyformer consistently outperforms state-of-the-art methods in experimental evaluations on public benchmark datasets, showcasing its effectiveness in capturing extensive temporal dependencies in dynamic graphs.
2: The integration of Message-Passing Neural Networks and Transformers in Todyformer offers a unique approach to addressing issues like over-squashing and over-smoothing in Temporal Graph Neural Networks.
3: Todyformer introduces novel strategies such as patchifying paradigm, structure-aware parametric tokenization, and temporal positional encoding, enhancing its ability to model evolving structural and temporal patterns.
4: The encoding architecture of Todyformer dynamically alternates between local and global contextualization, striking a balance that mitigates over-smoothing and improves model expressiveness.
5: Todyformer demonstrates significant performance gains on extra-large datasets, indicating its capability to scale up to data domains with extensive long-range interactions.

Cons of Todyformer:
1: The paper mentions that the hyperparameter search has not been exhausted during experimental evaluation, indicating that further optimization may be required to fine-tune the performance of Todyformer.

2: While Todyformer outperforms the two latest state-of-the-art methods, GraphMixer and Dygformer, the paper does not provide a detailed comparison with a broader range of existing models, potentially limiting the scope of the evaluation. A thorough literature review is required for a high quality paper. For example, the following two papers are highly relevant and using similar strategies, thus should be discussed and compared:
Transformer-Style Relational Reasoning with Dynamic Memory Updating for Temporal Network Modeling. AAAI'21.
Spatio-Temporal Attentive RNN for Node Classification in Temporal Attributed Graphs. IJCAI'19.
Besides, the following paper is also relevant, should be discussed:
Adaptive Neural Network for Node Classification in Dynamic Networks. ICDM'19.
Refer to the "3.5 Improving Over-Smoothing by Alternating Architectur". The following paper is highly relevant, thus should be discussed:
Learning to Drop: Robust Graph Neural Network via Topological Denoising. WSDM'21.

3: The paper acknowledges that no model exhibits consistent improvement across all datasets due to data imbalance issues inherent in anomaly detection tasks, suggesting that Todyformer may face challenges in handling imbalanced datasets.

Overall, Todyformer presents a promising approach to dynamic graph analysis, offering innovative solutions to common challenges in Temporal Graph Neural Networks and demonstrating strong performance in experimental evaluations.

---

> ### Comment · Reviewer_5WPT · 2024-04-11
> **what's the response?**
>
> Dear Authors,
>
> I did not see any responses and changes on request.
>
> best,

---

> ### Author Response · Authors · 2024-04-11
> **First Response to Reviewer1**
>
> Thanks for providing the positive feedback of Todyformer. The answers to the cons is as follows:
>
> weakness1: We acknowledge that our exploration of hyperparameters (HP) was not an exhaustive search, primarily due to constraints in computational resources and time, particularly given the extensive number of datasets involved. Additionally, we experimentally revealed the performance improvement of Todyformer over SoTA through the given HP search which indicates the strength of novelties to generalize to various datasets. Nevertheless, our current HP search yielded satisfactory results, as evidenced by the superior performance of the Todyformer model compared to the SOTA under these conditions. While we recognize the HP search was limited to major hyper-parameters within a limited range of values, we emphasize the success achieved in outperforming existing approaches through this straightforward approach. Additionally, on TGBL datasets, it is extremely difficult to extend HP search given our computation and time resources.
>
> weakness2: We thank the reviewer for bringing the related papers to our attention. The aforementioned papers have been cited in the related works section to help improve the literature review, where they were qualitatively discussed in comparison to existing baselines and our proposed architecture.
>
> weakness3: Thank you for bringing up this insightful observation. We've delved into the evaluation of our model's performance in handling imbalanced FLP task, as presented in Table 2 and Table 11. Here, we juxtaposed the test and validation performance of our model with the SOTA, focusing on MRR metric. The datasets and the evaluation protocol utilized were sourced from the TGBL paper [1], specifically tailored for imbalanced setups featuring multiple negative samples.  for instance in Flight or Coin and Comment the number of positive to negative samples is 1:100 for instance.  Our model underwent training and validation procedures based on a single positive sample alongside multiple negative samples.
>
> Requested changes:
> 1) We appreciate the paper recommendations. The papers you suggested have been referenced and deliberated upon in the related work section.
> 2) As previously stated, our model's performance underwent assessment using the TGBL leaderboard datasets, characterized by the inclusion of multiple negative samples, thereby introducing an imbalance in the evaluation protocol for link prediction tasks. As illustrated in 11, Todyformer outperforms the baseline models such as TGN, Dyrep and GraphMixer across various datasets such as TGBCOin, TGBComment, TGBFlight, and TGBWiki. Furthermore, it secures the second-highest position for TGBReview in terms of validation AP. Considering the time constraints and resource limitations, we conducted experiments on the FLP model using three toy datasets within an imbalanced setup, wherein each positive sample is accompanied by five negative samples. The provided table summarizes the findings of this initial investigation. Notably, our model demonstrates superior performance compared to MPNN-based dynamic graph encoders in terms of validation AP under these conditions. We anticipate expanding upon these datasets and models in subsequent discussions.
> | dataset|      Todyformer |  dyg2vec |
> |----------|:-------------:|------:|
> | Enron | 0.9486 | 0.9222 |
> | MOOC | 0.9487  | 0.9362   |
> | UCI |0.9493 |  0.9270   |
> [1] Huang, Shenyang, et al. "Temporal graph benchmark for machine learning on temporal graphs." Advances in Neural Information Processing Systems 36 (2024).

---

### Review · Reviewer_eUmK · 2024-03-06

**Summary Of Contributions:**

This paper introduces the Todyformer, a novel approach to modeling dynamic graphs through a three-stage process: patchifying, tokenization, and encoding. Unlike prior efforts that mostly address static graph structures, Todyformer uniquely targets dynamic graphs, where each edge includes a temporal dimension. The proposed method addresses the over-squashing issue prevalent in dynamic graphs by segmenting large graphs into smaller subgraphs during the patchifying stage, akin to techniques used in image processing. Additionally, Todyformer employs a dynamic graph neural network (DyG2Vec) in its tokenization process to effectively capture temporal patterns. The novel encoding stage combines both local and global encoders within a single layer, processing them sequentially to enhance model performance significantly.

**Audience:**

No

**Claims And Evidence:**

No

**Requested Changes:**

1. The methodology behind the ablation study on the alternating mode is unclear. Clarification is needed on whether the study compares architectures with sequential local and global encoders without node packing and unpacking processes. Detailed explanations are essential for understanding the impact of alternating modes on performance enhancement.
2. The term "window sizes" introduced in Section 4.3 requires a definition, as it appears without prior explanation in the methodology section.
3. Enhance readability by standardizing citation formatting. Enclose all citations within parentheses, for example, converting "Divakaran & Mohan (2020)" to "(Divakaran & Mohan, 2020)".
4. Figures 1 and 2 could benefit from improvements for better clarity:
    - Consider rotating textual elements in Figures 1 and 2 where feasible, such as $P_k$ and $n_k$, to facilitate easier reading.
    - Clarify the symbols used in the figures, specifying the representations of $n$, $P$, and $c$, and ensuring each symbol's meaning is easily accessible without extensive manuscript review.
    - Expand on the patchifying process within Figure 1, explicitly indicating whether each temporal subgraph corresponds to a patch and providing more detailed context.
5. Suggestions for notation improvements:
    - Reevaluate the use of "Tokenizer" for a module that functions similarly to an MPNN, focusing on message passing between nodes. A more intuitive naming convention might be to rename "Tokenizer" to simply "MPNN".
    - Replace "Transformer" with "Self-Attention" in Equation 6 to avoid confusion, as "Transformer" generally refers to a combination of self-attention and MLP layers.
    - Provide explicit details and notations for the PACKER and UNPACKER components, as their descriptions are currently lacking.

**Strengths And Weaknesses:**

Strengths:

- A primary novelty of Todyformer is its alternating use of local and global encoders within each layer. This approach addresses the challenge of handling dynamic graphs, which typically feature nodes with extensive neighboring connections compared to static graphs. This approach represents a key contribution in the field of dynamic graph representation.
- Experimental results demonstrate Todyformer's superior performance over existing baselines. The significance of the model's design choices is further validated through comprehensive ablation studies presented in Section 4.3.

Weaknesses:

- **The paper's presentation requires refinement.** The current use of terminologies and notations is often confusing, detracting from the overall readability. Additionally, the omission of crucial details undermines the paper's comprehensiveness. Refer to the "Requested Changes" section for more detailed feedback.
- **The novelty of the approach may be overstated.** While the paper asserts novel contributions in patch generation, structure-aware tokenization, and the utilization of Transformers to aggregate global context in an alternating fashion, some of these elements resemble concepts previously explored in works such as Dygformer. As such, the truly novel aspect appears to be the specific implementation of altering architecture with local and global layers, casting some doubt on the claimed extent of novelty.
- **The marginal performance gains observed in the Todyformer, relative to state-of-the-art architectures, seem to primarily result from the addition of more layers over DyG2Vec.** This raises questions about the actual efficacy of the proposed idea of altering local and global encoders and its substantial role in enhancing performance.
    - Although Todyformer marginally surpasses DyG2Vec across several datasets, as indicated in Table 1 and Figure 4, it incurs a significant increase in inference time—between two to four times longer than DyG2Vec.
    - This trade-off is somewhat justified by Todyformer's use of DyG2Vec for structure-aware tokenization within its processing pipeline.  Nonetheless, it suggests that the performance improvements attributed to the model's additional components might be marginal, achieved at the expense of higher computational resources.
    - The extent to which these improvements represent a meaningful advancement in the field hinges on a careful consideration of their practical impact versus the additional resources required.

---

> ### Author Response · Authors · 2024-04-11
> **First Response to Reviewer2**
>
> Thank you for offering positive and constructive feedback. We have carefully reviewed each comment and provided detailed responses below, taking them into account.
>
> Weakness1: Appreciation is extended for the nuanced and thorough constructive feedback. The requested adjustments have been meticulously addressed individually within the Requested Changes section.
>
> Weakness2: Apologies for any confusion. We strongly believe that the originality of our work has been overlooked. Although there may be minor similarities, such as the patchifying mechanism within Dygformer at the first sight, our model significantly enhances generalizability and expressive power through the following unique contributions. Moreover, to the best of our knowledge, we are the pioneer in proposing an approach to tackle long-range dependencies in continuous-time dynamic graphs using a graph/vision transformer paradigm through the following contributions:
> a) A structure-aware parametric tokenization strategy leveraging MPNNs to help overcome the expressive limitation of local encoders beyond their over-smoothing boundaries.
> b) A Transformer with temporal positional-encoding to capture long-range dependencies given higher-over node embeddings returned from the local encoding network.
> c) An encoding architecture that alternates between local and global contextualization, mitigating over-smoothing in MPNNs.
> d) A novel utilization of the patchifying mechanism to mitigate over-squashing in MPNN models tailored for dynamic graphs where the node degree is distributed over multiple dynamic sub-graphs across all of the patches.
> To highlight the advancements of our work relative to Dygformer, we have outlined additional novel features incorporated beyond the Dygformer model.
>
>  In the dygformer framework, patchifying entails the application of historical interaction analysis to a specific node's context. In otherwords, the interaction history of a node within the one-hop context is retrieved and encoded using Transformers. In contrast, our methodology involves the extraction  of subgraphs and subsequent higher-order message-passing within a patch, without strict emphasis on individual one-hop node neighborhoods. Consequently, our approach to patchifying and tokenization operates at the graph level where the input to Transformers is the encoded node embedding returned from the local encoding. This is diverging from dygformer's node-level approach. This process ensures that the embeddings generated from MPNN are updated utilizing message-passing information from all nodes within the corresponding subgraphs.
>
> Moreover, our architecture allows for the consideration of long-range structural dependencies among nodes situated within the same temporal range. We prioritize multi-hop interactions to generate embeddings for nodes, thereby enhancing the model's ability to capture complex relationships over extended distances within the temporal domain.
>
> In our approach, transformers are applied to the node embeddings computed by the tokenizer across various patches. This stands in contrast to dygformer, where transformers are applied to manually extracted features. Specifically, our model applies transformers to the outputs (M outputs) of tokenizers, ensuring the efficient and direct exchange of information across embeddings of the same node situated in temporally distant patches.
>
> Our model marks the pioneering introduction of an alternating mode designed to seamlessly transition between local and global information propagation within dynamic graph settings, achieving this with minimal complexity in the hope of improving the generalization by improving the boundary on over-smoothing.
>
> Weakness3: It is noteworthy to highlight that the performance achieved by todyformer is substantial, particularly evident when assessing large datasets featured on the TGBL leaderboard. Compared to other methods, todyformer demonstrates noteworthy improvements in Mean Reciprocal Rank (MRR), showcasing significant advantages. Notably, datasets like TGBLFlight, TGBLCoin, and TGBLComment, characterized by imbalanced labels, represent particularly challenging tasks for models, making the achievements of todyformer even more remarkable. Additionally, a Figure 7 has been included in the appendix, depicting experiments conducted with a large number of layers. Notably, the figure illustrates that the alternating mode results in a slight performance increase as the number of layers grows. Conversely, stacking layers of Message Passing Neural Networks (MPNNs) leads to a drop in model performance. Further details regarding the setup of these experiments and their analysis have been discussed in the first Review's Response.

---

> ### Author Response · Authors · 2024-04-11
> **Second Response to Reviewer2**
>
> Weakness4: Firstly, in Appendix A.5, we extensively examine the time complexity of Todyformer. Our analysis demonstrates that the complexity of our model closely aligns with that of MPNN-based dynamic graph encoders. However, it is imperative to note that our primary objective centered on improving the performance of Todyformer. Consequently, certain computational overheads are incurred in the packing and unpacking stages of the pipeline during the implementation. Nevertheless, these aspects can be optimized for efficiency in future extensions of Todyformer.
>
> Weakness5: Theoretically, our model exhibits the potential to achieve peak performance levels comparable to dyg2vec, particularly when considering large dynamic graphs. We acknowledge this observation and express our intention to explore this pathway for further investigation into the role of graph transformer architectures in enhancing the efficiency and effectiveness of learned representations for downstream tasks. While our experimental evaluations shed some light on the underlying factors contributing to our empirical achievements, due to constraints such as limited resources and time, we defer a detailed scrutiny of these aspects to future endeavors.
> Weakness6: As previously mentioned, we have deferred the optimization of performance versus efficiency to future extension works. Our primary focus has been on enhancing the expressive capabilities of dynamic graph encoders to capture long-range dependencies while maintaining qualitative time complexity, as demonstrated and compared in Appendix A.5.
>
> Requested Changes:
> 1) In Table 4, which presents the ablation study, we meticulously assess the impact of various components of the model architecture, particularly those associated with the alternating mode, such as packing and unpacking. This section has been comprehensively updated in the paper's appendix. The table delineates different scenarios:
> a) If the model lacks any of the three components (G.E, P.E, Alt.3), it signifies that the model solely relies on an MPNN-based encoder (local encoder).
> b) If it includes only G.E, it indicates the presence of packing alongside one block of Local and Global encoders without any unpacking.
> c) If G.E and P.E are present without Alt.3, this suggests the inclusion of one global and local encoder block coupled with patch positional encoding.
> d)Finally, if all components are present, it signifies the incorporation of three blocks of local and global encoders with positional encoding, as well as packing and unpacking modules within each block.
>
> 2)  We introduced W as the window size in the Problem Formulation (Section 3) and also included it in the Table of Notations in the Appendix section. However, to enhance the readability of the paper and improve the traceability of the notations, we additionally illustrated the window size in an example in Figure 1.
>
> 3) As per the TMLR LaTeX template instructions, there are two types of citation formats: when the authors or the publication are included in the sentence, the citation should not be in parentheses. we have updated all the citation accordingly.
> 4) Thank you for your comment. We applied the requested changes and explained the notations used in Figure 1 directly into the caption.  As for Figure 2, the notations remain largely the same, with the exception of the 'c' symbols, which we have opted out to remove for the sake of simplicity and redundancy elimination. We have made improvements to Figure 1, particularly in clarifying the patchifying process. Specifically, we have depicted how the entire graph is divided into M subgraphs (patches) with an equal number of edges in each patch, based on the timestamps of the edges. Additionally, we have updated the caption to provide comprehensive details on the patchifying process.
>
> 5) Thank you for highlighting these aspects. We have maintained our alignment with graph/vision transformers, hence opting in for the term "Tokenizer," which can encompass any graph encoder, such as MPNN. For consistency, we have changed the tokenizer's name into local encoder in the paper. We utilized the Transformer Encoder to compute contextualized representations for each node in the graph, leveraging the representations learned for the same node across multiple patches. It's worth noting that our approach extends beyond simple Self-Attention, encompassing the full architecture of the Transformer. However, we recognize that the explanation provided in section 3.4 Global Encoding (Transformer) may be somewhat incomplete, potentially causing ambiguity. To address this concern, we have enhanced the explanation and explicitly outlined the steps taken to provide clearer insights within the paper.

---

### Review · Reviewer_jrhC · 2024-03-27

**Summary Of Contributions:**

The paper introduces Todyformer, a Transformer-based neural network tailored for dynamic graphs, used for Temporal Graph Neural Networks. Todyformer unifies local encoding capabilities of Message-Passing Neural Networks (MPNNs) with the global encoding prowess of Transformers.

**Audience:**

Yes

**Claims And Evidence:**

No

**Requested Changes:**

Refer to Weaknesses

**Strengths And Weaknesses:**

Strengths:

1. the overall performance of Todyformer looks quite good.

Weaknesses:

1. the notation is not clear, especially for eq(6). H^l is a tensor of shape N×M×D, why the shape of Q, K, V is NxD? How M is eliminated?
2. the figure 1 is also not clear, the shape is NxM or NxMxD?
3. it seems the computational cost is very large, it could be (NxM)^2.
4. the patch partitioning is not clear, how the time is considered here? just evenly split them? Besides, how the Window size W is considered here?
5. The motivation of this paper is to "improve model expressiveness through alleviating over-squashing and over- smoothing in a systematic manner." However, the experiment results seems cannot support this motivation. There are not clear evidence that the proposed method can alleviate over-squashing and over-smoothing.

---

> ### Author Response · Authors · 2024-04-11
> **First Response to Reviewer1**
>
> Thank you for acknowledging the positive and constructive feedback. Additionally, we have addressed and clarified some of the ambiguous points as indicated below.
>
> Weakness1: Apologies for any confusion and typographical errors in the text. The correct size for the matrices Q, K, and V should be MxD which is resolved in Eq6.  This implies that attention is computed across the embeddings of similar node derived from different patches. the attention parameters (Q, K, V) are shared across different nodes.
>
> Weakness2: thanks for raising it. We have included Figure 5 and its corresponding description in Appendix A.3.1 to illustrate the relationship between input and output tensors, along with their respective notations. Moreover, Figure 1 has been revised to present a schematic representation illustrating the procedure of selecting different embeddings of a node across multiple patches. It also depicts how a sequence is generated for each node and subsequently fed into the global encoder individually.
>
> Weakness3:  As we mentioned in the architecture, the attention is calculated across the patches. In other words, since the transfomer is applied the embeddings of each node derived from different patches, the maximum complexity of transformer is M^2. As for each of N nodes the transformer is called once, the final complexity will be Nx(M^2).  we elaborated more on complexity analysis in  section A.5.1 which details time complexities.
>
> Weakness4: The sorted edge list, organized by timestamps, is evenly divided into M patches to ensure uniformity in the number of edges within each patch. The window size dictates the receptive field of our architecture, corresponding to the time-frame under scrutiny in graph analysis. In dynamic graph learning literature [1], it is a prevalent strategy to select a substantial subgraph as the input graph, independent of the network's remaining structure. This facilitates the partitioning of large dynamic graphs, enabling inductive and transductive data separation for training and testing purposes.  Thus, our experimental protocol setup aligns with baseline methods [2]. Without loss of generality, and for notation brevity, we assume that \mathcal{G}_{ij}​ within window size W is implicit from the context and serves as the input graph. Finally, having the input graph extracted given a window size, the patchifying module evenly partition the input graph into subgraph such that they each have equal number of edges.
>
> Weakness5: Thank you for bringing up this point. It's important to note that the primary contribution of this paper is enhancing the expressive power of dynamic transformers to capture long-range dependencies in both time and structure. Our motivation on the over-smoothing and over-squashing is speculated supported by emprical results and architecture designs. We indeed need to theoretically investigate this emprical finding and support it with theoretical analysis. Having this taken into consideration, we have updated the abstract and introduction sections to accurately convey this message.
> Given the time constraints of this round of discussion, we conducted experiments to showcase the robustness of our model against over-smoothing. The results are presented in the table below and further elaborated upon in Figures 6 and 7 of the paper. In these new experiments, we explored a larger number of layers for both MPNN and Todyformer, comparing their validation AP. We observed that as the number of layers increases, the performance of MPNN decreases, whereas Todyformer's performance shows slight improvement.
> It's worth noting that to accommodate the larger models and fit them into GPUs, we conducted these experiments on smaller datasets (Enron) and utilized neighbor sampling with a size of one for the layers after the first layer. This neighbor sampling strategy  prevent significant drops in performance of MPNNs, although the overall decreasing pattern remains consistent.
> Further discussion on oversmoothing and oversquashing has been detailed in Appendix A.6 of the paper. The result of Enron dataset is as follows:
> | model|      Num Layers     |  AP (%)|
> |----------|:-------------:|------:|
> | dyg2vec | 20 | 98.24   |
> | dyg2vec | 30 |  98.05  |
> | dyg2vec | 40 |  97.52  |
> | dyg2vec | 50 |  97.28  |
> | Todyformer | 20 | 98.95   |
> | Todyformer | 30 |  99.05  |
> | Todyformer | 40 |  99.11 |
> | Todyformer | 50 |  99.28  |

---

### Decision · Action_Editor_eEuS · 2024-05-27

**Recommendation:** Accept with minor revision

**Comment:**

The paper has proposed a new method of Todyformer for modeling dynamic graphs, and provided experiments to demonstrate its superior performance. The contribution is solid and will be beneficial to the TMLR community. The reviewers have provided further suggestions in their recommendations, based on which the authors could further revise the paper. Below, I summarize those comments.

1. Please further polish the paper to improve the presentation.
2. Some of the proposed methods resemble concepts previously explored in works such as Dygformer. Please further clarify the differences of the proposed method from the previous approaches.
3. The performance gains observed in the Todyformer, relative to state-of-the-art architectures, seem to primarily result from adding more layers over DyG2Vec. Please clarify if this is the case.

**Audience:**

Yes. The topic will be interesting to a large set of TMLR audience.

**Claims And Evidence:**

Yes. The authors have provided experiments that demonstrate the desired performance of their proposed method.